# Nuclear actin structure regulates chromatin accessibility

Buer Sen[1], Zhihui Xie[1], Michelle D. Thomas[2], Samantha G. Pattenden[2], Sean Howard [3], Cody McGrath[1], Maya Styner [1], Gunes Uzer [3], Terrence S. Furey [4] & Janet Rubin [1] ✉

Polymerized β-actin may provide a structural basis for chromatin accessibility and actin transport into the nucleus can guide mesenchymal stem cell (MSC) differentiation. Using MSC, we show that using CK666 to inhibit Arp2/3 directed secondary actin branching results in decreased nuclear actin structure, and significantly alters chromatin access measured with ATACseq at 24 h. The ATAC-seq results due to CK666 are distinct from those caused by cytochalasin D (CytoD), which enhances nuclear actin structure. In addition, nuclear visualization shows Arp2/3 inhibition decreases pericentric H3K9me3 marks. CytoD, alternatively, induces redistribution of H3K27me3 marks centrally. Such alterations in chromatin landscape are consistent with differential gene expression associated with distinctive differentiation patterns. Further, knockdown of the non-enzymatic monomeric actin binding protein, Arp4, leads to extensive chromatin unpacking, but only a modest increase in transcription, indicating an active role for actin-Arp4 in transcription. These data indicate that dynamic actin remodeling can regulate chromatin interactions.

Remodeling of the cellular actin cytoskeleton contributes to dynamic changes in cell phenotype and function[1,2]. Recently, changes in cytoplasmic actin structure were discovered to affect transport of actin into the nucleus[3–5] where nuclear actin is essential for proper gene transcription and DNA repair[6,7]. In stem cells, the intranuclear state of actin with regard to the ratio of monomeric (G-actin) to polymeric (F-actin) has been shown to guide differentiative pathways[8]. Moreover, nuclear G-actin interactions with actin-related-proteins (e.g., Arp4 and Arp8) regulate the transcription of genes within heterochromatin[9], but little is known regarding how intranuclear actin structures affect accessible chromatin to drive gene transcription.

The nucleus includes a nucleoskeleton made up of A/C and B type lamins that provide scaffolding for heterochromatin[10] and thus organization[11]. Once a cell reaches its terminal differentiated state, the lamin A/C nucleoskeleton appears to be fixed until deregulated with aging or cancer[12]. In contrast to the largely static lamin network, the cellular actin cytoskeleton continuously remodels. For instance, when cells stiffen in response to extracellular matrix cues, polymerized cytoplasmic F-actin generates force on the nucleus through connections to Linker of Nucleoskeleton and Cytoskeleton (LINC) complexes, allowing heterochromatin to be rearranged on lamin scaffolds[13]. Actin is also present within the nucleus[14], and its energy-dependent import and export are regulated by extracellular processes including mechanical strain[3,4,6]. Due to the density of the nucleus, F-actin actin rarely appears as strongly phalloidin-positive strands typical of cytoplasmic F-actin, although it can be visualized as distinct actin fibers in some states such as DNA replication stress[6,15], or transiently during mitotic exit[16]. Despite poor geographic visualization of polymerized actin, evidence suggests that a dynamic exchange between free and polymerized actin states occurs in the nucleus, and importantly, most actin polymerization modifiers, including Arps, have been found within the nucleus[14,17]. The fact that actin polymers can be assembled

[1]Department of Medicine, University of North Carolina, Chapel Hill, NC, USA. [2]Division of Chemical Biology and Medicinal Chemistry, Center for Integrative Chemical Biology and Drug Discovery, UNC Eshelman School of Pharmacy, University of North Carolina at Chapel Hill, Chapel Hill, NC, USA. [3]Department of Mechanical and Biomedical Engineering, Boise State University, Boise, ID, USA. [4]Departments of Genetics and Biology, University of North Carolina at Chapel Hill, Chapel Hill, NC, USA. ✉e-mail: jrubin@med.unc.edu

and disassembled within the nucleus, and that actin has been shown to regulate chromatin remodeling complexes[18], begs the question of the role of intranuclear actin in modulating the epigenetic nuclear landscape.

β-actin itself is required for proper functioning of the Brahma-associated factor (BAF) complex and specifically associates with BRG1 subunit of BAF[19]. The BAF complex plays critical roles in differentiation and development through regulation of chromatin around key genes, in part by affecting the activity of polycomb repressive complexes. Knock down of β-actin affects chromatin organization, both in terms of accessibility as well as intra-chromatin interactions and with histones[7]. A recent crystal structure of human BAF bound to a nucleosome[20] showed that BAF contains an actin-related protein (Arp) complex consisting of SMARCA4, β-actin, and Arp4 (ACTL6A/BAF53A)[19]. This complex links the BAF ATPase module, which directly interacts with the nucleosome and the BAF base module; thus changes in actin associations would be predicted to have effects at regulating chromatin states.

Stem cells undergo profound changes in their gene transcription profiles during the transition from stemness/ multi-potentiality into specified terminal identities, during which process a large change in chromatin accessibility occurs and constitutive chromatin repressive marks are irreversibly altered[21]. The multipotential bone marrow mesenchymal stem cell (MSC), which we study here, differentiates to supply precursors for osteoblasts which form bone, and marrow adipocytes which store energy for local use[22]. By virtue of micro-environmental controls specific to bone marrow, bone MSC are primed to enter the osteoprogenitor lineage, such that a switch into adipocyte lineage requires un-silencing and expression of a non-osteoblastic gene network[23]. One way to alter these transitions is by changing actin structure in the nucleus: CytoD, for example, depolymerizes cytoplasmic F-actin, inducing transport of G-actin monomers into the nucleus from which CytoD is excluded, and thus nuclear actin structure is increased; the increased nuclear actin augments maturation into osteoblasts[5]. In contrast, blocking Arp2/3 action to increase secondary actin branching (using the enzymatic inhibitor CK666) provokes a lineage switch into the adipogenic pathway[8]. Arp2/3 complexes are located at membrane surfaces - in the nucleus presumably at the inner nuclear membrane - where the Arps can interact with both lamins and constitutive heterochromatin[24]. Additionally, Arp4, an actin related protein that has no enzymatic activity, is found only in the nucleus where it has been shown to inhibit F-actin polymerization[25] and function as a critical member in chromatin modifying complexes[26]. Arp4's effect on differentiation through actin remodeling has not been studied. This information suggests that modulation of actin content and structure in the nucleus might regulate access to gene regulatory elements and subsequently modify gene transcription during stem cell differentiation.

Our goal in this work was to understand if remodeling nuclear β-actin structure would alter DNA accessibility due to treatment with cytoskeletal inhibitors CK666 and CytoD. CK666 prevents Arp2/3 induced secondary actin branching at 70° from end on end polymers[27], and CytoD increases actin transport into the nucleus[5]. These actin states induce discrete cell phenotype responses: CK666 causes a lineage switch to adipogenesis, and CytoD promotes osteoblastogenesis in the absence of other differentiative cues[8,28]. We describe changes in cell morphology, nuclear organization including heterochromatin location, and gene expression. Presented data shows that imposition of sustained structural changes in β-actin leads to specific alterations in chromatin accessibility and the geographic location of both facultative and constitutive heterochromatin marks. Our work shows that the epigenetic landscape of MSCs is subject to control by actin remodeling.

## Results

### Nuclear actin states and connectivity with cytoskeleton are modified by CK666 and Cytochalasin D

We previously showed that two commonly used actin modifiers, the Arp2/3 inhibitor CK666 and CytoD, an inhibitor of cytoplasmic actin polymerization, alter the differentiation program of MSCs which emerge as adipocytes or mature osteoblasts, respectively, after 3 days in culture[5,8,28]. In both cases, actin within the nucleus is objectively altered. CK666 prevents secondary branching, shown as a nearly complete absence of phalloidin staining within the nucleus 3 h after treatment (Fig. 1A, B). The findings shown throughout have been repeated in at least 3 × 6-well experiments with a high power image from each experimental well examined. Supplementary Fig. 1a shows examples of the fields from which cells are selected in a non-biased fashion; cells representing each treatment condition (control, CK666, CytoD) are easily identifiable by the cellular actin staining pattern. Intranuclear actin state is further demonstrated by transfecting with the NA chromobody (Nuclear Actin Chromobody-TagGFP plasmid) in Fig. 1C. The NA chromobody stain reveals scant, but recognizable actin fibrils in the control condition, which are increased with CytoD treatment and decreased in the CK666 condition. (Six further examples of imaged cells transfected with the NA chromobody are shown in Supplementary Fig. 1b) Arp3 remains both intra- and extra-nuclear, its location undisturbed by inhibition with CK666 (Fig. 1D). Preventing secondary branching with CK666, despite similar total actin within the nucleus, decreases nuclear F-actin with a consequent rise in G/F actin ratio, shown in a representative blot in Fig. 1E, and accompanied by densitometry representing 5 separate experiments. Quantitative densitometry confirms that treatment with CK666 decreases F-actin within the nucleus ($p < 0.0001$).

In contrast, an inhibitor of actin polymerization, CytoD, that fails to enter the nucleus due to lack of endogenous transporter mechanisms[8,29], promotes both actin transport into the nucleus and an increase in nuclear F-actin, perturbing lamin A/C lining the inner nuclear membrane (Fig. 1A, B). Visible points of phalloidin staining are consistent with polymerized structures charted within the nucleus[15,30]. After CytoD, Airy magnification of the nuclear area shows polymeric actin adherent to the outer nuclear membrane forming a thick cap both above and below stained lamin A/C on cross section. Further, CytoD treated cells stained with the nuclear actin chromobody show clear appearance of F-actin fibrils, as shown in Fig. 1C and Supplementary Fig. 1b. Densitometry for nuclear actin (Fig. 1E) reveals statistically similar levels of F-actin and G-actin in nuclei of control MSCs, and large increases in F-actin after treatment with CytoD.

Examination of cytoplasmic F-actin structure shows that focal adhesions at the membrane as well as around the nucleus are decreased after CK666, resulting in decreased actin cables traversing the nucleus (Fig. 2A, stained with vinculin) and reduce filapodial actin accretions. CytoD prevents formation and maturation of focal adhesions in the cytoplasm[1], which results in actin points or globules instead of the arcing cables seen in control and CK666 treated cells. Interestingly, Arp3 relocates to this extranuclear actin structure, and may be important to the generation of the multiple actin polymers visualized there (Fig. 1D). Combining cytoplasmic actin visualization with the confocal images and quantitation of increased F-actin in Fig. 1, our data indicate that CytoD, but not CK666, is associated with enhanced actin polymerization in the nucleus.

Measurements of cell modulus largely represent the stiffness of its largest and densest organelle, the nucleus. The modulus of CK666 treated cells is not significantly different from control cells, while CytoD decreases cell modulus despite the presence of actin around the nucleus (Fig. 2B). Decreased modulus apparent with cytochalasin D

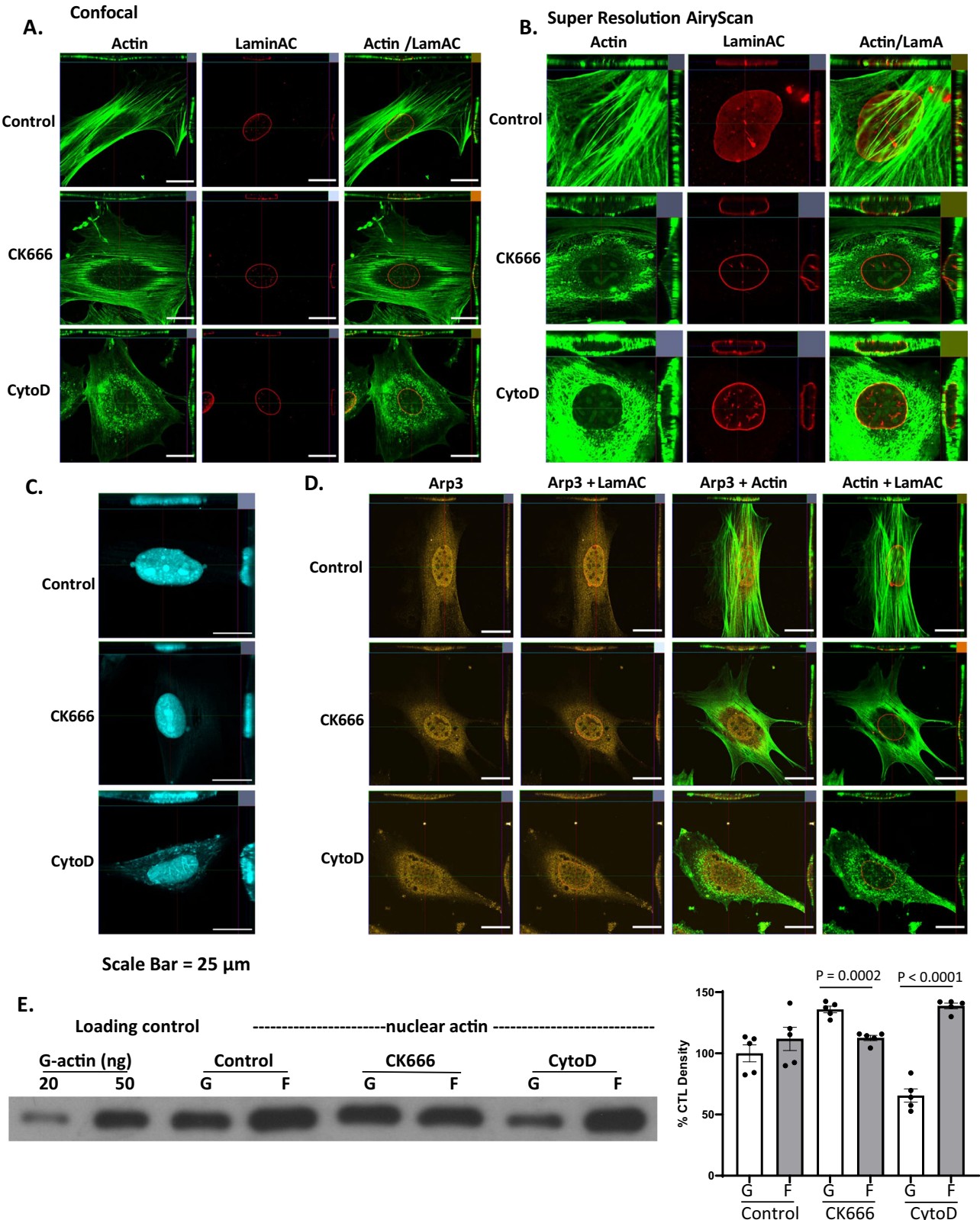

Scale Bar = 25 μm

could be in part due to the reduced number of F-actin cables traversing the apical nuclear surface, which are known to contribute to the atomic force microscope measured stiffness[31]. The decreased modulus due to CytoD treatment is further decreased in the presence of CK666: this finding suggests that the presence of structural actin within the nucleus does supply some portion of the measured cell and nuclear modulus.

## Nuclear actin state alters chromatin accessibility and gene transcription

Remodeling actin structure within and around the nucleus induces profound effects on chromatin accessibility and transcription. We performed ATAC-seq to determine chromatin accessibility profiles across perturbations in actin structure caused by CK666 and CytoD at 24 h. A principal components analysis (PCA) indicates that CK666

**Fig. 1 | Actin structure In the nucleus is altered by CK666 and CytoD.** Mouse marrow-derived mesenchymal stem cell (MSC) were treated with CK666 (100 μM) or CytoD (0.1 μg/ml) for 4 h. **A, B** The cells were stained for Lamin A/C (red) and F-actin (green, phalloidin) and then examined using a model LSM 880 confocal microscope (Zeiss, Thornwood, NY) and (**B**) visualized using Airyscan. **C** Cells were transfected with the nuclear actin chromobody and imaged 4 h after treatment as specified. **D** The cells were stained for Arp3 (yellow, cy5), Lamin A/C (red) and F-actin (green, phalloidin). For images, a random cell in each of 6 HPF was selected

for imaging, and representative image shown; scale bar: 25 μm. Imaging was performed in 3 separate experiments. **E** Nuclear G- and F-actin immunoblot analysis using G/F actin assay kit (Cytoskeleton, Inc.) for control, CK666 and Cyto D-treated MSCs. Data points represent mean densitometry ±SEM relative to vehicle-treated samples for $n = 5$ biologically independent samples; two way ANOVA with Sidak post-hoc adjusted for multiple comparison,; $p = 0.0002$ for CK666 and <0.0001 for CytoD. Source data are provided as a Source Data file.

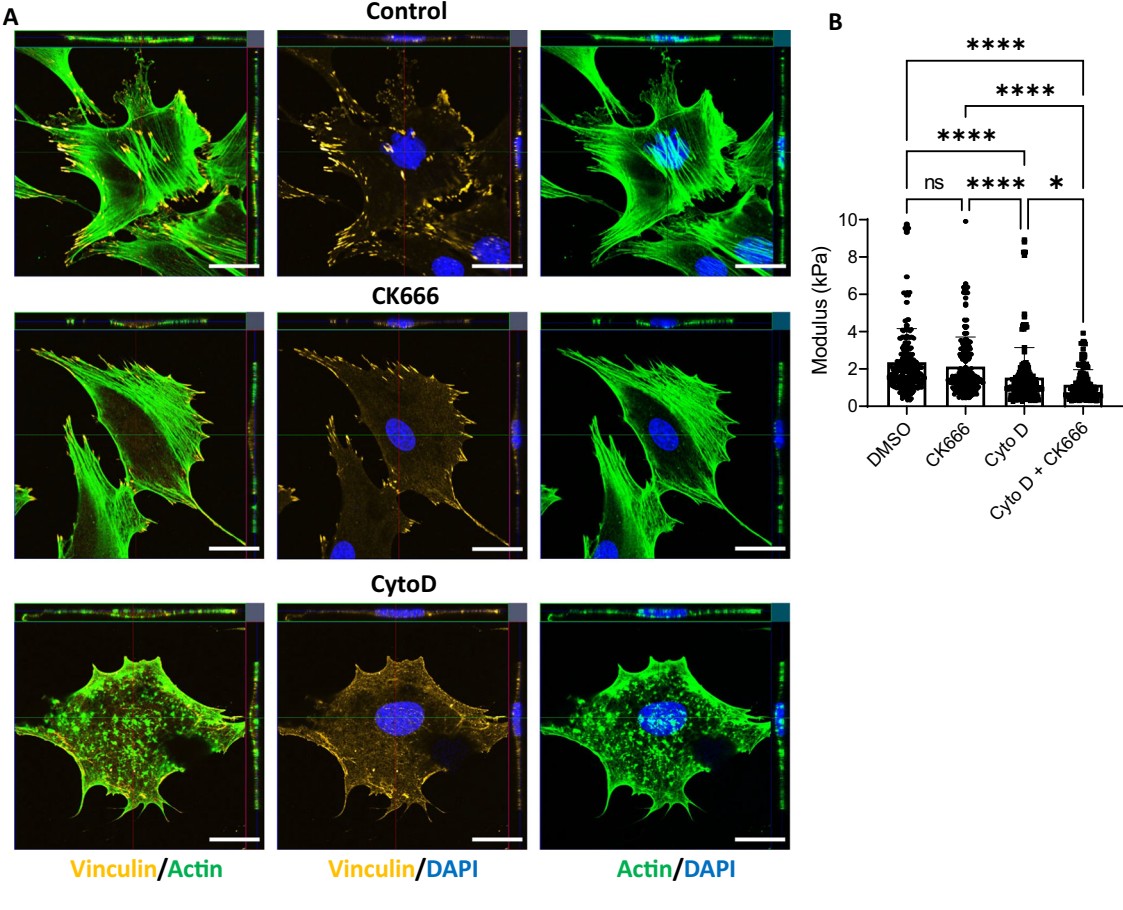

**Fig. 2 | Actin connections to nucleus are altered by CK666 and CytoD. A** MSC were treated with CK666 or CytoD same as Fig. 1, and then stained focal adhesion (yellow, cy5), F-actin (green; phalloidin), and nucleus (blue; NucBlue). Images are representative of at least 3 biological repeats; scale bar: 25 μm. **B** Measurement of

cell modulus, data for each condition acquired from 5 measures per each of 60 independent cells and presented as mean ± SEM; one way ANOVA with Tukey post hoc test adjusted for multiple comparisons; between group differences are shown as ****$p < 0.0001$, *$p < 0.05$. Source data are provided as a Source Data file.

creates greater alterations in the overall chromatin landscape than does CytoD (Fig. 3A). We performed differential chromatin accessibility analysis to determine specifically how chromatin was changing across the two conditions compared to control. For CK666, we found more regions of significantly increased accessibility with greater magnitudes of change compared to those with decreased accessibility (Fig. 3B; $p < 0.01$; $\log_{10}$ fold change >2; Supplementary Data 1). CytoD treatment, alternatively, resulted in more regions with decreased chromatin accessibility (Fig. 3C). Of the 7471 (4476 increased, 2995 decreased) regions significantly altered by CK666 and the 6113 (2102 increased, 4011 decreased) altered by CytoD, 1157 (360 increased, 797 decreased; Supplementary Data 2) were significantly altered in the same direction in both conditions (Fig. 3D), representing less than 20% of the total in each.

To assess how chromatin accessibility patterns affected regulation of transcription, we measured gene expression using RNA-seq.

Again, PCA (Fig. 4A) shows treatment with nuclear actin modifiers creates statistically significant changes in expression at 24 h with the overall expression profile most affected by CK666. Compared with chromatin accessibility, we did not find as much difference in the number and magnitude of increased and decreased gene expression for either CK666 (Fig. 4B) or CytoD (Fig. 4C). We found more differentially expressed genes in common between the two conditions (376 total, 164 up-regulated, 212 down-regulated) with 24% of the 1583 differential genes (710 up-regulated, 873 down-regulated; Supplementary Data 3) altered by CK666 and 30% of the 1256 differential genes (665 up-regulated, 591 down-regulated; Supplementary Data 4) in this overlap (Fig. 4D). Most of the transcriptional changes were uniquely ascribed to a specific condition. It is useful to note that MSC do not assume 'terminal' phenotypes until several days later[23], and early changes up to 24 h likely reflect operational starts to lineage acquisition[32] rather than terminal identities. As well, CytoD, although

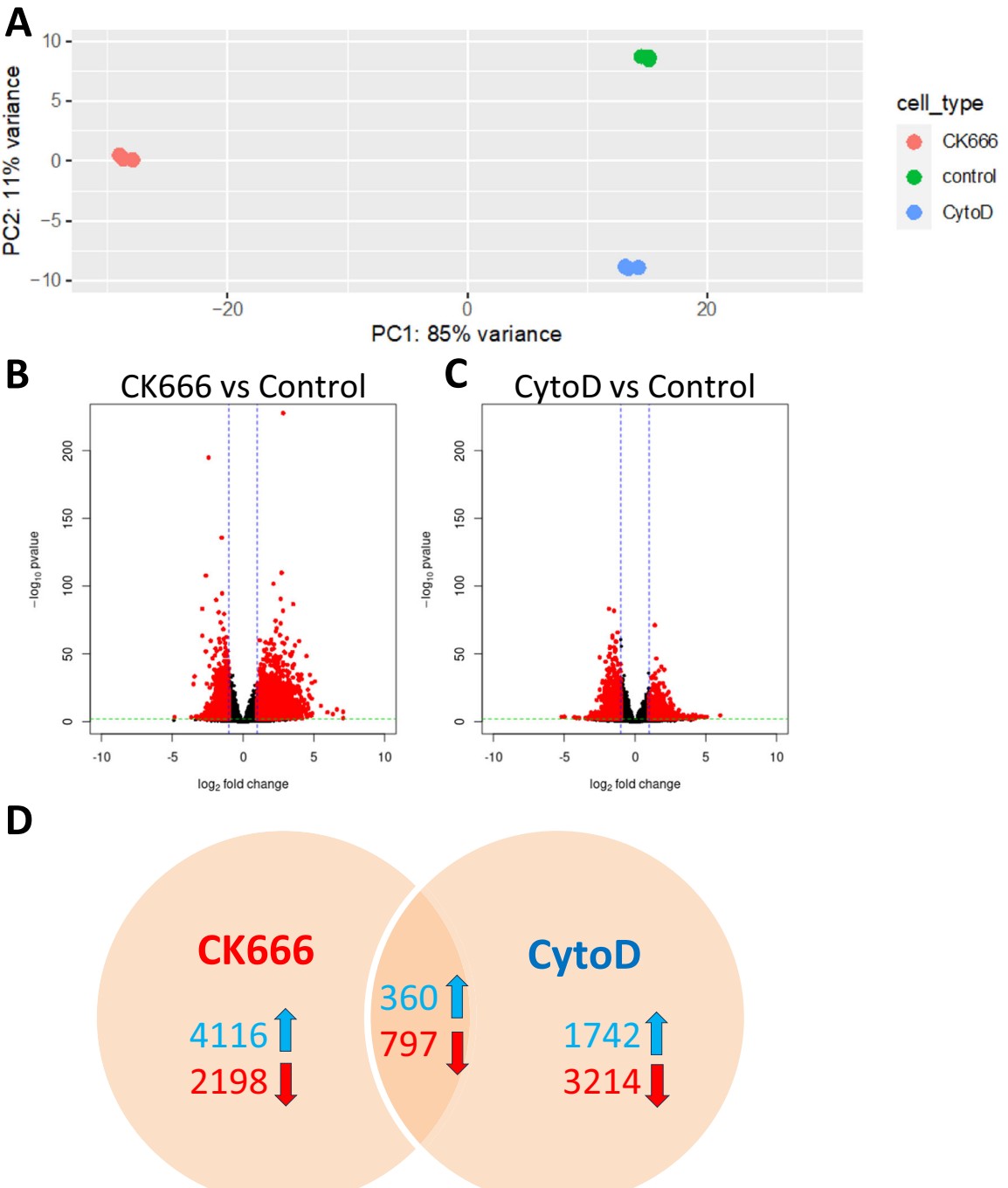

**Fig. 3 | Chromatin accessibility altered by CK666 and CytoD. A** PCA shows CK666 (red) alters chromatin accessibility relatively more than CytoD (blue) when compared to the baseline control state (green). **B** Addition of CK666 results in more regions with significantly ($p < 0.01$; DESeq2 Wald test; adjusted using Benjamini-Hochberg correction) increased accessibility with greater magnitudes than regions with decreased accessibility. **C** Addition of CytoD results in more regions with significantly ($p < 0.01$; DESeq2 Wald test; adjusted using Benjamini-Hochberg correction) decreased accessibility. **D** Most chromatin changes are unique to a given condition (85% of CK666 regions; 81% of CytoD regions).

largely promoting a terminal osteoblast phenotype in MSC, also induces adipogenesis in some cells in the exposed population[5].

To better understand those cellular processes most affected by CK666 and CytoD, we performed overrepresentation analysis (ORA) on both genes linked to differentially accessible chromatin and differentially expressed genes (Fig. 4E). We performed separate ORA on genes with increased and decreased accessibility or expression. Pathways were manually clustered based on their enrichment patterns across the two conditions. As expected, genes related to adipogenesis

showed increased chromatin accessibility and expression in cells exposed to CK666[8], suggesting changes leading to adipocyte differentiation are present at even 24 h (see Supplementary Fig. 2 for 3 day effect in these experiments). Interestingly, early adipogenesis pathways were also activated with CytoD, but as we reported in vitro and in vivo[5] and show again in Supplementary Fig. 2, the predominant differentiation by 3 days after CytoD treatment is osteoblastogenesis. This data is further supported by heat-maps in Supplementary Fig. 3, which show that CytoD increases baseline osteogenic genes by 24 h,

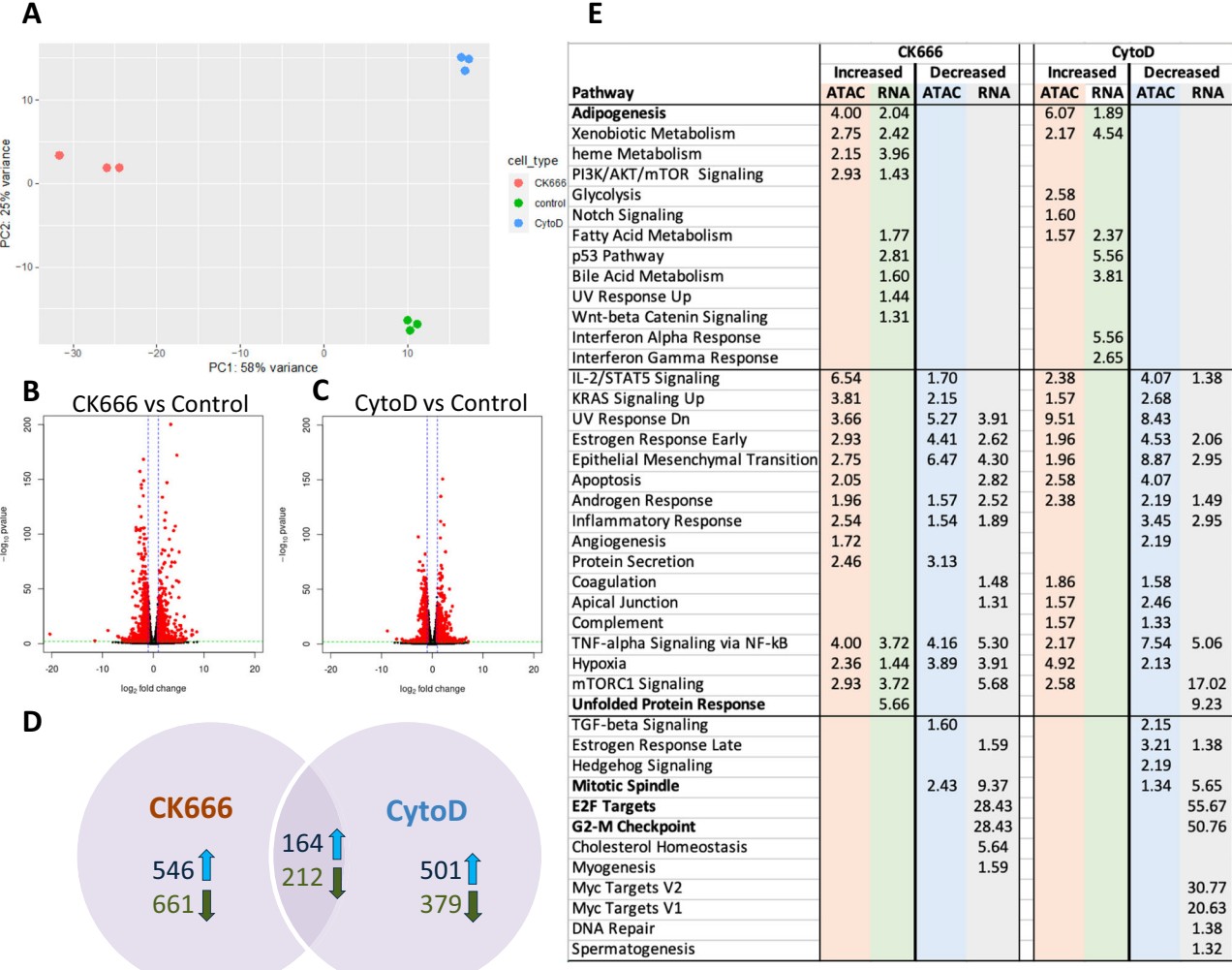

| Pathway | CK666 Increased ATAC | CK666 Increased RNA | CK666 Decreased ATAC | CK666 Decreased RNA | CytoD Increased ATAC | CytoD Increased RNA | CytoD Decreased ATAC | CytoD Decreased RNA |
|---|---|---|---|---|---|---|---|---|
| **Adipogenesis** | 4.00 | 2.04 | | | 6.07 | 1.89 | | |
| Xenobiotic Metabolism | 2.75 | 2.42 | | | 2.17 | 4.54 | | |
| heme Metabolism | 2.15 | 3.96 | | | | | | |
| PI3K/AKT/mTOR Signaling | 2.93 | 1.43 | | | | | | |
| Glycolysis | | | | | 2.58 | | | |
| Notch Signaling | | | | | 1.60 | | | |
| Fatty Acid Metabolism | | 1.77 | | | 1.57 | 2.37 | | |
| p53 Pathway | | 2.81 | | | | 5.56 | | |
| Bile Acid Metabolism | | 1.60 | | | | 3.81 | | |
| UV Response Up | | 1.44 | | | | | | |
| Wnt-beta Catenin Signaling | | 1.31 | | | | | | |
| Interferon Alpha Response | | | | | | 5.56 | | |
| Interferon Gamma Response | | | | | | 2.65 | | |
| IL-2/STAT5 Signaling | 6.54 | | 1.70 | | 2.38 | | 4.07 | 1.38 |
| KRAS Signaling Up | 3.81 | | 2.15 | | 1.57 | | 2.68 | |
| UV Response Dn | 3.66 | | 5.27 | 3.91 | 9.51 | | 8.43 | |
| Estrogen Response Early | 2.93 | | 4.41 | 2.62 | 1.96 | | 4.53 | 2.06 |
| Epithelial Mesenchymal Transition | 2.75 | | 6.47 | 4.30 | 1.96 | | 8.87 | 2.95 |
| Apoptosis | 2.05 | | 2.82 | | 2.58 | | 4.07 | |
| Androgen Response | 1.96 | | 1.57 | 2.52 | 2.38 | | 2.19 | 1.49 |
| Inflammatory Response | 2.54 | | 1.54 | 1.89 | | | 3.45 | 2.95 |
| Angiogenesis | 1.72 | | | | | | 2.19 | |
| Protein Secretion | 2.46 | | 3.13 | | | | | |
| Coagulation | | | | 1.48 | 1.86 | | 1.58 | |
| Apical Junction | | | | 1.31 | 1.57 | | 2.46 | |
| Complement | | | | | 1.57 | | 1.33 | |
| TNF-alpha Signaling via NF-kB | 4.00 | 3.72 | 4.16 | 5.30 | 2.17 | | 7.54 | 5.06 |
| Hypoxia | 2.36 | 1.44 | 3.89 | 3.91 | 4.92 | | 2.13 | |
| mTORC1 Signaling | 2.93 | 3.72 | | 5.68 | 2.58 | | | 17.02 |
| **Unfolded Protein Response** | | 5.66 | | | | | | 9.23 |
| TGF-beta Signaling | | | 1.60 | | | | 2.15 | |
| Estrogen Response Late | | | | 1.59 | | | 3.21 | 1.38 |
| Hedgehog Signaling | | | | | | | 2.19 | |
| **Mitotic Spindle** | | | 2.43 | 9.37 | | | 1.34 | 5.65 |
| **E2F Targets** | | | | 28.43 | | | | 55.67 |
| **G2-M Checkpoint** | | | | 28.43 | | | | 50.76 |
| Cholesterol Homeostasis | | | | 5.64 | | | | |
| Myogenesis | | | | 1.59 | | | | |
| Myc Targets V2 | | | | | | | | 30.77 |
| Myc Targets V1 | | | | | | | | 20.63 |
| DNA Repair | | | | | | | | 1.38 |
| Spermatogenesis | | | | | | | | 1.32 |

**Fig. 4 | Gene expression altered relatively less than chromatin by CK666 and CytoD. A** PCA shows CK666 (red) alters gene expression relatively more than CytoD (blue) when compared to the baseline control state (green), similar to chromatin accessibility. **B** Addition of CK666 results in similar numbers of genes with significantly (*p* < 0.01; DESeq2 Wald test; adjusted using Benjamini-Hochberg correction) increased and decreased expression and of similar magnitudes. **C** Addition of CytoD also results in similar numbers of genes with significantly (*p* < 0.01; DESeq2 Wald test; adjusted using Benjamini-Hochberg correction) increased and decreased expression, but with lower magnitudes than CK666. **D** The number of differentially expressed genes is less than altered chromatin regions, and relatively more are shared between conditions (24% of CK666 genes; 30% of CytoD genes). **E** Pathway enrichment for genes near differentially accessible chromatin (ATAC) and differentially expressed genes (RNA). Values are −log₁₀ (adjusted *p* value of enrichment). See also Supplementary Fig. 1 which shows that CK666 induces adipogenesis at 3 days, while CytoD induces an osteoblast phenotype.

while CK666 does not; CytoD also increases adipogenic genes, but this pattern is largely replaced by osteogenesis at 3–4 days. Shown in Supplementary Fig. 4, where standard medias were used to terminally differentiate osteoblasts and adipocytes[5], different actin cytoskeletons sort to these phenotypes: actin structure is depleted in adipocytes and increased in osteoblasts. Nuclear actin and Arp4 staining are not visibly different in any of the differentiated nuclei compared with control cells. This suggests that significant changes in nuclear actin and chromatin states caused by actin remodeling drugs, while initiating differentiating events, eventually achieve some adaptive equilibrium.

The most significantly enriched pathways common to both actin-altering conditions were decreased expression of genes targeted by the E2F transcription factors and involved in the G2 to M phase checkpoint in the cell cycle. Chromatin accessibility for cell cycle associated regulatory elements was not similarly altered. These suggest a significant reduction in cell cycle activity indicating reduced growth rates, potentially due to differentiation away from the self-renewing MSC state. Many pathways showed significant changes in chromatin accessibility in both directions simultaneously, most with concomitant decreases in gene expression. Most of these are signaling-related pathways and may reflect that cellular stress increases in the presence of significantly altered actin dynamics.

Most changes in chromatin and expression were either consistent in direction across both conditions or were only found in one condition. A notable exception is the Unfolded Protein Response (UPR) pathway that was found to be significantly upregulated by CK666 while being significantly downregulated by CytoD. The UPR pathway relieves endoplasmic reticulum stress and plays an important role in many cellular processes, including differentiation of MSCs[33]. UPR pathway up-regulation due to CK666 might reflect the stress of MSC switching pathways from an orientation favoring osteoblast development, while its downregulation with CytoD may be consistent with greater retention of the molecular characteristics of the pre-osteoblast control state[23]. In sum, inducing actin modifications across the cell generates a state which forces the MSC out of the multipotential state, shutting down proliferation genes, and invoking adaptive responses leading to differentiation.

We also examined CK666 induced changes in chromatin accessibility at 4 h to complement the 24 h data set. We found that while changes in chromatin accessibility again clearly separated CK666 samples from control in a PCA (Supplementary Fig. 5a), there were fewer altered regions (1962 total, 1098 increased, 864 decreased; Supplementary Fig. 5b). Of these, 774 were shared at 4 h and 24 h (Supplementary Fig. 5c), with associated genes enriched in common pathways, notably adipogenesis (Supplementary Fig. 5d, Supplementary Data 9), suggestive of a progressive mechanism whereby chromatin is modified.

To provide a contrast in a different cellular setting, we measured chromatin accessibility in the well-studied NIH-3T3 embryonic fibroblast cell line, previously used to investigate intranuclear actin structure[34,35]. Here, we found that 4 h after addition of CK666, treated and control cells were clearly separated by PCA (Supplementary Fig. 6a): chromatin accessibility was significantly altered in 8091 regions (4827 increased, 3264 decreased; Supplementary Fig. 6b). Comparing to MSCs at both 4 and 24 h post CK666 treatment, most of these changes were unique to the NIH 3T3 cells (Supplementary Fig. 6c). PCA of MSCs and NIH 3T3 cells together show that chromatin profiles at baseline are extremely different, as expected for functionally distinct cells; this variation far exceeded the variation induced by CK666 (Supplementary Fig. 6d). Interestingly, pathways enriched by genes near altered chromatin in NIH 3T3 cells largely overlapped those enriched in MSCs, though there was not a significant signal for adipogenesis (Supplementary Data 10). These data suggest that specific regions altered by inhibition of secondary actin branching depend on the baseline cell chromatin state, but that there may be common cellular processes affected by these changes.

## Discrete alterations in location of histone marked heterochromatin are ascribed to different intranuclear actin states

Hetero- and euchromatin associations with the lamin nucleoskeleton become largely stable at the same time that lamin A/C is expressed in the MSC[36]. A full array of actin modifying enzymes found in the nucleus indicates that the actin polymers could offer scaffolds upon which chromatin could be arrayed during differentiation. We thus explored whether preventing secondary branching within the nucleus would change the structural organization of heterochromatin marks, thus changing chromatin accessibility as confirmed by our ATACseq data (Fig. 3). Changes in the "constitutive" heterochromatin mark, H3K9me3, are associated with maintenance of cell identity and lineage stability[37,38]. We found that CK666, but not CytoD, caused significant alterations in H3K9me3 mark location, which aggregate in foci ("K9-foci") (Fig. 5A, B). Such H3K9me3 foci have been described as "distinct, multi-chromosomal, membrane-less heterochromatin domains" where H3K9me3 is associated with HP1[39]. Four hours after exposure to CK666, the nuclear area was expectedly smaller due to less prominent apical fibers exerting less tension on the nucleus, allowing it to bulge upward; however, the density of H3K9me3 was unchanged, as were total levels measured by densitometry (Fig. 5B). Importantly, the number of K9-foci significantly decreased ($19.5 \pm 3$ in CK666 condition compared to $30.3 \pm 4$ in control, difference $p < 0.0001$), with mobilization toward the inner nuclear membrane, shown in the far right graph of Fig. 5A. In comparison, treatment with CytoD decreased the number of K9-foci to a far lesser degree ($25 \pm 4$ for CytoD compared to $30.3 \pm 4$ in control, $p < 0.001$). Importantly, although both CK666 and CytoD increase the amount of nuclear actin (Fig. 1D), the formation of both primary and branched actin polymers can continue and even increase due to increased substrate in nuclei of cells treated with CytoD, which is excluded from the nucleus[5]. These data support the hypothesis that intranuclear actin provides a modifiable structure for chromatin arrangements.

We next examined the facultative heterochromatin mark, H3K27me3, previously shown to be downregulated during osteogenic differentiation due to treatment with CytoD[40]. Figure 5C shows that in control and CK666 cells, H3K27me3, visible diffusely across the nucleus, was concentrated at the inner nuclear membrane. CytoD caused a quantitatively significant relocation away from the nuclear membrane toward the center of the cell (Fig. 5D, $p < 0.0001$). No treatment caused measurable changes in total H3K27me3 compared with control (Fig. 5E). In sum, alterations in nuclear actin, whether via increasing nuclear actin with cytochalasin D, or increasing free actin through limiting secondary branching via CK666 inhibition of Arp2/3, differentially causes heterochromatin relocalization in the absence of changing total amounts of H3K9 or H3K27 trimethylation.

## Arp4 interactions with actin are critical to chromatin accessibility and gene transcription

Formation of nuclear F-actin decreases the G-actin pool[41], and we here showed that the corollary, i.e., inhibition of Arp2/3 actin polymerization, generates the opposite effect: CK666 increases the ratio of G- to F-actin in the nucleus (Fig. 1D). This finding stimulated a consideration of effects on chromatin accessibility due to the actin-related protein Arp4 that forms a heteromeric complex with actin monomers[19,42]. While Arp4 is not known to have enzymatic activity, its binding of monomeric actin decreases substrate to form F-actin, and thus its knockdown leads to an increase in F-actin[25]. Arp4 is also known to interact with HP1-alpha[43], which has a role in H3K9me3 deposition and localization[25].

We first confirmed that marrow derived MSC expressed Arp4, which we found was entirely confined to the nucleus (Fig. 6A). Treatment with CK666 induced a movement of Arp4, presumably bound to monomeric actin, towards the edges of the nucleus with increased distribution along folded lamin of the taller and less spread nucleus as shown in the figure. To investigate whether the relocation of H3K9me3 from pericentric heterochromatin toward the nuclear edge involved Arp4, we measured the location of this methyl mark after CK666 treatment. As before, CK666 caused a significant decrease in the number of K9-heterochromatin marked foci in a pattern similar to Fig. 5A (Fig. 6A, B). Knock-down of Arp4 was consistent with a trend toward decreased numbers of heterochromatin marked foci, but this steady state was not further decreased by CK666 (Fig. 6C). This suggests that when actin monomers are released through CK666 inhibition, extant Arp4 enables the K9-foci loss from the central nucleus.

Shown in Fig. 6D (and Supplementary Fig. 2), CK666 treated cells generated Adiponectin and Ap2 proteins at 3 days during transition to an adipogenic phenotype. Expression of these fat markers was inhibited when Arp4 was knocked down. Interestingly, adipogenesis due to "adipogenic medium", widely used to induce adipogenesis[23], was also repressed in the absence of Arp4 (Fig. 6E). This finding suggests that the Arp4/actin associations might control both accessibility and transcription of adipogenic genes during MSC differentiation.

Next, we determined the effect of Arp4 knockdown on the chromatin landscape and expression profile of these cells. We found that chromatin accessibility was affected to a much greater extent by Arp4 knockdown than by addition of CK666, with variation from the control state primarily due to the absence of Arp4 regardless of presence of CK666 (Fig. 7A). More specifically, we found 37,158 chromatin regions affected by Arp4 knockdown alone, dominated in number and magnitude by increased accessibility (Fig. 7B; 31674 increased, 5484 decreased; Supplementary Data 5). When actin branching was prevented by CK666, leading to increased G-actin in the nucleus, chromatin accessibility changes largely reflected those that due to loss of Arp4 (Fig. 7C, D). Of the 32,559 chromatin alterations (25,575 increased, 6984 decreased; Supplementary Data 6), 23196 (71%) were similar to those in Arp4 knockdown only, including nearly 80% of the regions with increased accessibility.

The effect on gene expression was not as extreme. Variability in the overall expression profile from the control state was still primarily

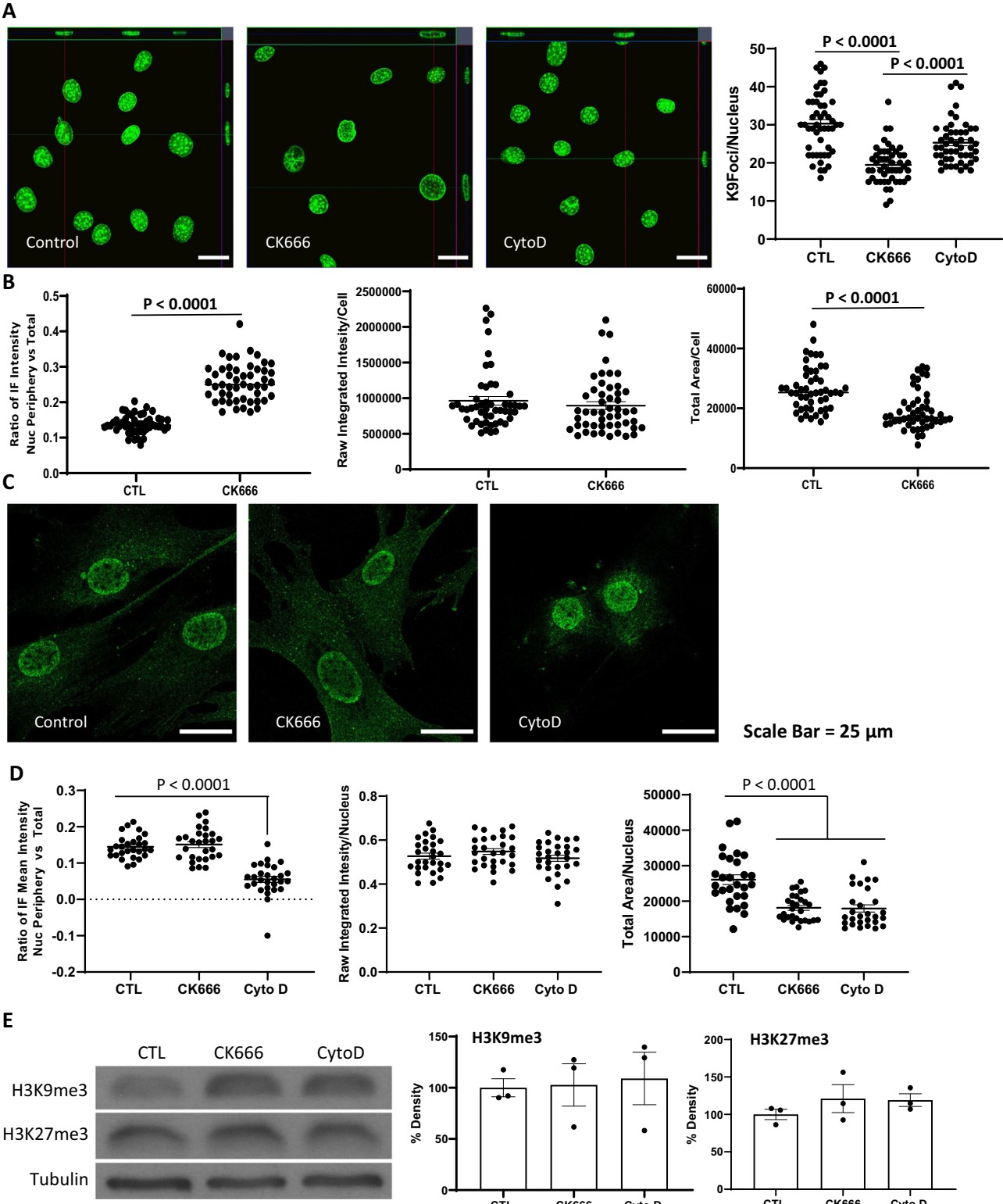

**Fig. 5 | Histone mark locations change with alterations in nuclear actin structure.** The cells were treated with/without CK666 or CytoD for 4 h. **A** The cells stained for H3K9me3 and foci counted for heterochromatin called as "K9foci", data for each condition was acquired from 50 independent cells and presented as mean ± SEM; one way ANOVA with Tukey post hoc test adjusted for multiple comparisons, between group differences are shown if $p < 0.0001$. **B** K9foci redistributed with CK666 to inner nuclear region; data acquired from 50 independent cells for each condition and presented as mean ± SEM, two-tailed t-test with $p < 0.0001$. **C** The cells were stained for H3K27me3. **D** H3K9me3 distribution in nucleus, data for each condition was acquired from 28 independent cells and presented as mean ± SEM; one way ANOVA with Tukey post hoc test adjusted for multiple comparisons, between group differences are shown if $p < 0.0001$. **E** Total H3K9me3 and H3K9me3 protein level was measured by Western blot; data points represent mean densitometry, relative to vehicle-treated samples for $n = 3$ biologically independent samples are presented as mean ± SEM, between group differences were not significant. Source data are provided as a Source Data file.

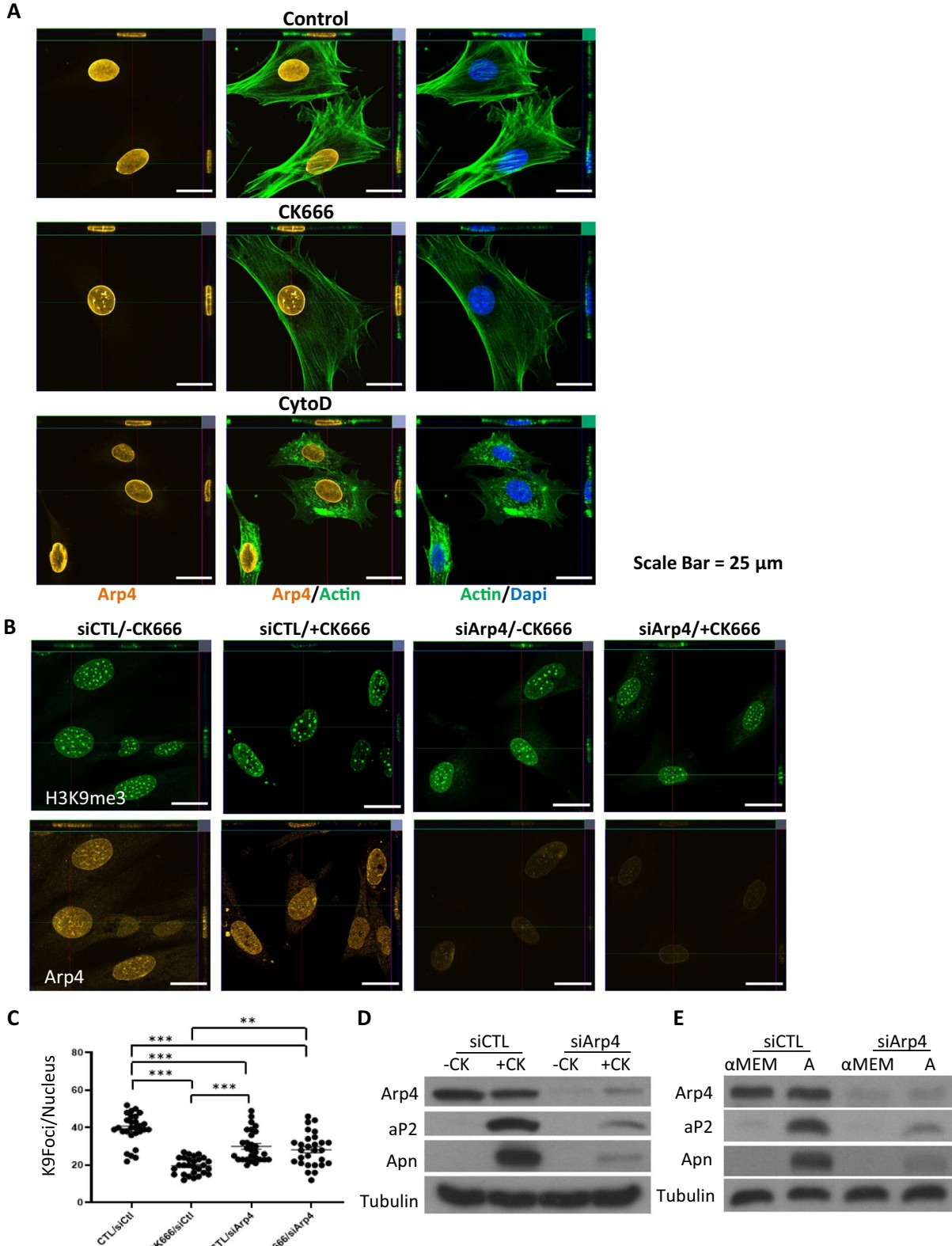

**Fig. 6 | Knockdown of Arp4 changes nuclear structure and prevents adipogenesis.** The cells were stained for Arp4 (yellow, cy5), F-actin (green, phalloidin) and nucleus (blue; NucBlue). **A** The cells were treated with/without CK666 or CytoD for 4 h. **B** Arp4 was knocked-down and treated with/without CK666. For images (**A**, **B**), 6 HPF were imaged, and representative image shown; scale bar: 25 µm. Imaging was performed in 3 separate experiments. **C** K9foci counted; data for each condition was acquired from 28 independent cells and presented mean ± SEM; two way ANOVA with Tukey post hoc test adjusted for multiple comparisons shows group differences, ***$p$ < 0.0001, **$p$ = 0.003. **D**, **E** Arp4 was knocked-down and the cells treated with/without CK666 or A medium for 3 days. Western blot for Arp4, aP2 and Apn. Source data are provided as a Source Data file.

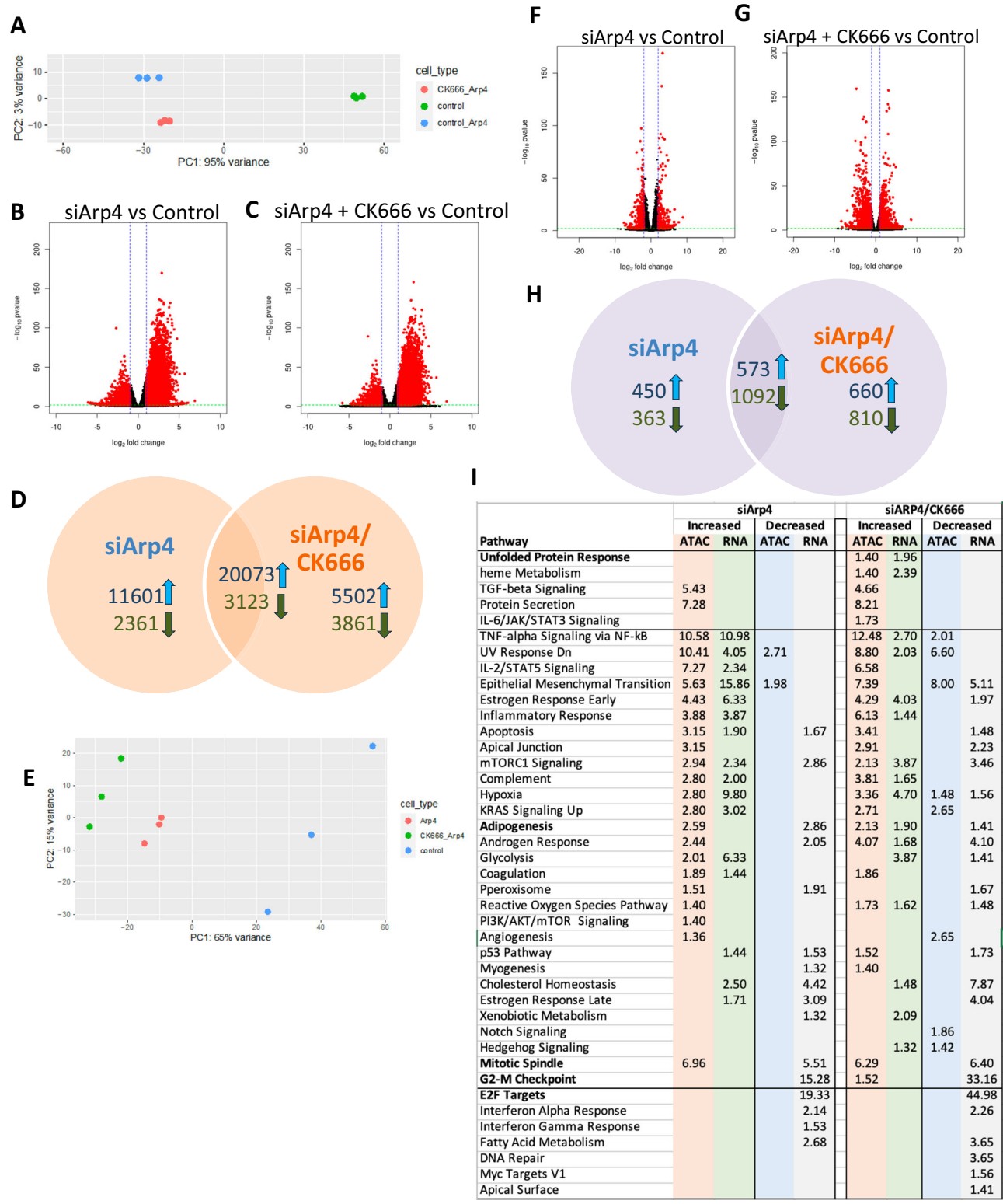

**Fig. 7 | Arp4 knockdown affects chromatin accessibility more than transcription. A** PCA of ATAC-seq data (**B**, **C**) Volcano plots of ATAC-seq data (DESeq2 Wald test; *p* values adjusted using Benjamini-Hochberg correction) (**D**) Overlaps of ATAC-seq data (**E**) PCA of RNA-seq data (**F**, **G**) Volcano plots of RNA-seq data (DESeq2 Wald test; *p* values adjusted using Benjamini-Hochberg correction) (**H**) Overlaps of RNA-seq data (**I**) Combined pathway analysis.

due to Arp4 knockdown (Fig. 7E), and indeed a larger total number of genes were affected by Arp4 knockdown (2478; 1023 increased, 1455 decreased; Fig. 7F; Supplementary Data 7) compared to addition of CK666 alone (1583; Fig. 3B). We did see more variability in controls which were treated with the siRNA transfection protocol than shown in un-transfected replicates. Comparing the chromatin accessibility and gene expression changes, it is noteworthy that the 60% increase in differentially expressed genes is substantially lower than the nearly 500% increase in affected chromatin regions. Further, the combined knockdown of Arp4 and addition of CK666 led to a greater number

and magnitude of gene expression changes (3135; 1233 increased, 1902 decreased; Fig. 7G; Supplementary Data 8) with slightly fewer in common with Arp4 knockdown alone (1665/2478, 67%; Fig. 7H) compared with chromatin changes. Interestingly, in both experiments there were more genes with decreased rather than increased expression, despite chromatin accessibility being overwhelmingly increased. A pathway analysis (Fig. 7I) showed many similarities between cellular processes affected by Arp4 knockdown and those due to addition of CK666 alone, including the decreased expression of genes targeted by the E2F transcription factors and involved in the G2 to M phase checkpoint in the cell cycle. But there were notable differences, including a decreased expression of adipogenesis genes despite increased chromatin accessibility in their vicinity, and there was no enrichment in changes to chromatin or expression of genes in the UPR pathway. The further addition of CK666 did appear to restore some of the molecular changes in these latter pathways, but not to the extent seen with CK666 addition alone. Together, these results suggest that Arp4 along with free actin plays an important role in chromatin availability and the regulation of gene transcription.

## Discussion

Access to discrete gene regulatory elements in the genome profoundly changes as a stem cell moves from a proliferative stem state into terminal lineage. By virtue of its residence within the bone marrow, the marrow derived MSC is primed to act as a regenerative cell of the skeleton, becoming a bone osteoblast. The alternate differentiation pathway for this cell is adipocytic, a transition which requires a major shift in expressed genes, beginning with PPARγ and C/EBPα and involving shut down of the gene arrays associated with osteoblastogenesis[23]. In previous work we found that actin modifiers can direct selection between these pathways: preventing actin branching (CK666) induces adipogenesis[8] and increasing actin transport into the nucleus by disassembling cytoplasmic actin polymers (Cytochalasin D) robustly promotes osteogenesis[5]. This suggested that, alongside the nucleoskeleton scaffold for tertiary and quaternary chromatin, dynamic actin structures also contribute to chromatin architecture. In the work presented here, we now show that actin modifiers cause large shifts in the accessible DNA 24 h after their addition, reflected in ATAC-seq data sets (Figs. 3, 4, 7). Transcription, as shown by RNA-seq, mirror some of the changes in chromatin accessibility, supporting, in the case of CK666, an increase in adipogenesis and TNF-alpha signaling. Decreased chromatin accessibility after either increases or decreases in nuclear actin structure is seen near genes involved in a host of pathways, yet overall, RNA expression was less affected than chromatin accessibility at the 24 h time point. As might be expected early during the process of differentiation, both actin modifiers decreased mRNA gene pathways associated with proliferation; interestingly decreased chromatin accessibility did not appear to be a major driver of this effect.

A role for intra-nuclear actin—both actin level and arrangement—in controlling gene regulation and expression might be predicted as nearly all members of the actin toolbox and processes to closely regulate actin entrance and egress from the nucleus have been well described[44]. Moreover, actin based functions contribute to genome stability along with affecting global transcription patterns[45]. Indeed, the absence of nuclear actin due to knockdown of β-actin caused a general decrease in gene activation through associated chromatin remodeling[7]. Mahmood et al analyzed several aspects of chromatin state and showed that knocking down β-actin levels altered chromatin structure via chromatin remodelers such as BAF/ BRG1 and EZH2, the latter which we have also shown to be influenced by changes in actin structure[8,40]. As well, expression of progerin, the mutant lamin A involved in Hutchinson-Gilford progeria syndrome, also leads to decreased nuclear actin and gene expression, and can be partially corrected by overexpression of nuclear-targeted actin[46]. Insofar as the

state of actin, as we confirm in this work, increased nuclear actin promotes transition from the proliferative state and eventually constrains the non-proliferating cell into differentiative phenotypes; increasing the proportion of the branched F-actin form in the nucleus leads to osteoblast lineage while decreased nuclear F-actin leads to adipogenesis[47]. Other examples showing that nuclear G- to F-actin ratios are important in controlling cell function include that the presence of nuclear F-actin is critical to the ability of cancer cells to resist chemotherapy[48], and necessary to increase production of cytokines in T-cells[49]. Here, limiting the ability to form branched actin in the nucleus of marrow MSC cells, which are poised to enter the osteoblast lineage, forces the cell not only out of the proliferative state, but into an alternate adipogenic lineage. Not unexpectedly, treatment with CK666 led to distinct chromatin profiles compared to controls in NIH3T3 cells (Supplementary Fig. 6). The numbers of gene regions near altered chromatin at 4 h was higher than those in the MSC cells, but included similar pathways. This commonality may point to chromatin arrangements that are shared between cell types.

Previous work has evaluated transient changes in nuclear actin state and chromatin geography. F- actin fibers form during mitotic exit, a process which involves the interacting molecule cofilin-1 that both severs actin filaments and transports actin monomers into the nucleus[16]. The appearance of F-actin invokes chromatin decondensation and nuclear expansion. Reorganization of chromatin after mitosis requires the presence of α-actinin 4, the full process occurring over about 60 min[34]. Interestingly, changes in intracellular calcium, either due to treatment with an ionophore, or through G-protein coupled receptors, can induce rapid stimulation of nuclear actin polymerization that peaks at less than 2 min before returning to a basal state[35]. And, as noted above, knocking out nuclear β-actin, essentially decreasing any actin contribution to structure, leads to widespread changes in chromatin arrangement[7]. Our data complements and adds detail to the types of nuclear actin structure that can lead to changes in specific chromatin accessibility in both MSCs and NIH3T3 cells. The story certainly extends beyond the nucleus to cytoplasmic actin structure which apply force on the nucleus[50] as well as many other modifiable structural proteins (e.g., microtubules) that connect and interact with the LINC complex and thus to static structural proteins inside the nucleus (e.g., lamins). As actin structure within the nucleus is dependent both on transport into the nucleus requiring cofilin and importin-9[5,51], and can be enacted upon by an actin toolbox within the nucleus[52], the dynamic control of nuclear actin level and structure is certain to be a central mechanism in modifying the epigenetic landscape and subsequent gene expression.

During differentiation, genes that orchestrate lineage[53] or preserve stemness[54] exchange positions between the periphery and the nucleoplasm. Such chromatin changes are largely irreversible leading to a general decrease in accessible chromatin as stem cells differentiate[21]. Accompanying these early changes in chromatin access due to actin modifiers, we found alterations in both facultative and constitutive heterochromatin marks. Inhibition of Arp2/3 causes the facultative H3K9me3 mark to decrease in pericentric locations during early adipogenesis. When such H3K9me3 relocation was limited by knock down of Arp4, adipogenesis was prevented. The significant effect on the geographical location of these marks, but not the total nuclear density of marks (Fig. 4) might imply that a stable phase separation that allows for compaction in pericentric locations[39] was altered by preventing secondary actin formation. Methylated histone marks are generally associated with repressed chromatin, silencing lineage-inappropriate genes[37], and key to maintaining stemness. Interestingly, it was during the switch from expected osteoblastogenic outcome into an alternative lineage where location changes in the constitutive H3K9me3 mark were seen; this might suggest that branched actin helps to retain epigenetic geographical stasis after differentiation pathways are selected. In turn, after addition of CytoD, distribution of the facultative H3K27me3 marks

toward the central nucleus was associated with increased attainment of the osteogenic endpoint. These data support that actin remodeling alters the structural chromatin landscape modulating specific gene expression pathways.

Heterochromatin states in stem cells are affected by both F-actin generated forces on the nuclear surface and within the nucleus. Our data indicated that CK666 does not alter nuclear mechanics while H3K9me3 foci move to the periphery. This suggest that apical fibers and LINC complex connectivity throughout the cell remains intact in the presence of CK666 as release of LINC connectivity is known to reduce H3K9me3 peripheral enrichment[55] and the location of H3K9me3 foci remain relatively unchanged under LINC disruption[56]. As such, the change in location of H3K9me3 marks due to reduction in branched actin likely depends on nuclear actin structure. CytoD on the other hand decreases nuclear stiffness presumably due to loss of apical F-actin fibers that interact with LINC connections.

To identify mechanisms by which loss of actin structure caused the H3K9me3 associated DNA to move from a central position to the inner nuclear membrane (Fig. 4), we considered that CK666 treatment increased the percentage of G-actin in the nucleus (Fig. 1). Arp4 is known to depolymerize actin filaments[25], and appears to complex with monomeric actin[42]. We did not find any geographic differences in Arp4 during differentiation induced by actin modifying drugs, or in terminally differentiated cells (Supplementary Fig. 4): it is found throughout the nucleus, with a suggestion of greater content at the inner nuclear membrane. However, knock-down of Arp4, which leads to decreased compaction of chromatin in yeast[26], not only prevented localization of H3K9me3 marks to the periphery of the nucleus due to CK666, but also entirely blocked lineage switching of MSC to adipogenesis. Interestingly, research has shown that HP1α, which has a role in H3K9me3 deposition and localization[25], is also associated with Arp4[43]; thus it is possible that H3K9me3 location may depend on loss of HP1 associated heterochromatin domain stability[39]. The enormous instability induced by Arp4 knockdown is evidenced by unmasking of large portions of the genome was not accompanied by similar magnitude changes in gene expression (Fig. 7). Previously it was shown that knockdown of Arp4 did not change the machinery for actin import, and as well did not change expression of many genes[25]. Combined, this information would suggest that Arp4 has a direct role in allowing transcription and is perhaps key to the need for actin monomers for this process[3].

In sum, inducing actin modifications within the nucleus forces the MSC out of the resting multipotential state, shutting down proliferation genes, and invoking adaptive responses leading to differentiation. CytoD induction of actin transport into the nucleus, where it can be operated on by the nuclear toolbox, significantly changes chromatin accessibility and expression by 24 h, but in a trajectory that is closer to the original state of the bone MSC, as a majority of these cells eventually attain the mature osteoblast state[5]. Preventing secondary actin branching alters the cell trajectory in the most extreme fashion tested here, switching the osteoprogenitor outcome of the MSC into the adipogenic lineage. Large changes in chromatin unwinding due to remodeling of the nuclear actin landscape may make available a selection of previously unavailable genes as the cell samples adaptive mechanisms to survive under stress. How actin modifiers change gene expression with input such as physical force, replication stress or cell adhesion, all which alter cellular actin remodeling, is a subject for future studies.

## Methods
### Materials
Fetal bovine serum was from Atlanta Biologicals (Atlanta, GA). Culture media, trypsin-EDTA reagent, antibiotics from Sigma-Aldrich (St. Louis, MO). Ibidi USA μ-Slide chamber (cat#: NC0515977) from Fisher (Hampton, NH).

### Cells and culture conditions
Mouse marrow-derived MSCs were harvested from six C57BL/6J murine bone marrow using a validated protocol, (UNC IACUC approved 23-204.0). NIH3T3 cells were from ATCC. Cells were maintained in minimal essential medium containing 10% FBS, 100 μg/mL penicillin/streptomycin. Adipogenic medium consisted of α-MEM supplemented with 10% FBS, 100 U/ml penicillin, 100 μg/ml streptomycin, 5 μg/ml insulin, 0.1 μM dexamethasone and 50 μM indomethacin. Osteogenic medium consisted of α-MEM supplemented with 10% FBS, 100 U/ml penicillin, 100 μg/ml streptomycin, 50 ng/ml vitamin C and 20 mM β-glycerophosphate.

### RNA Interference
Cells were transfected with siRNA (25–50 nM) in serum-free OptiMEM overnight before replacing the medium and adding reagents for cell treatment, using the Life Technology protocol for Lipofectamine RNAi MAX (cat #13778150). RNA interference studies were performed with siRNAs for Arp4: 5′-CCUCUGGCUGGAGACUUCAUUACCA; siCTL for arp4: 5′-CCUGGUCGGGACAUUACUUAUCCCA.

### Immunofluorescence Microscopy
For microscopy, cells were seeded in U-slide four-well chamber (ibidi, Gräfelfing, Germany; cat# 80426). Cells were fixed with 4% paraformaldehyde for 10 min, permeabilized in 0.1% Triton-X 100 for 5 min, and blocked with 5% donkey serum for 30 min. Three consecutive washes for 10 min with phosphate-buffered saline (PBS) were performed between each step. Incubation with primary antibodies at 37 °C for 1 h, which are anti-Lam A/C (Abcam; cat# ab26300), anti-Arp3 (Invitrogen, PA5118867), anti-Actl6A (Arp4) (Sigma; cat# SAB4503429-100UG), anti-Vinculin (Sigma; cat# V4505-.2 ML), anti-H3K9me3 Abcam; cat# ab176916, anti-H3K27me3 (Abcam; cat# ab6002). For evaluation of nuclear filaments, cells were transfected with the Nuclear Actin Chromobody-TagGFP plasmid (pnAC-TagGFP, Chromotech, code: acg-n) using Lipofectamine 3000 (cat# L3000, Life Technologies) and cultured at 37 °C for 48 h prior to cell treatments. For all experiments a single cell per HPF was imaged, to include 6 cells from each condition. The selection of the cell was random, any from a field which was significantly changed by the addition of either CK666, CytoD (Supplementary Fig. 1a) and conditions with siRNA added.

Visualization of primary antibodies was performed with Rhodamine Red-conjugated anti-rabbit (cat# 711- 295-152), Cy5-conjugated Donkey Anti-Rabbit IgG (cat# 711-175-152), Alexa Fluor 488-conjugated Donkey Anti-Mouse IgG (cat# 715-545-151). These secondary antibodies were obtained from Jackson Immuno Research (West Grove, PA). Actin stress fibers were examined using Alexa Fluor 488-conjugated phalloidin (cat# A12379; Lifetechnologies), Nuclei were visualized by NucBlue reagent (Life Technologies; cat# R37605). After 3 consecutive 10-minute washes with PBS, the cells in chamber slides were covered with PBS. For 3D images, cells were examined using a model LSM 880 confocal microscope (Zeiss, Thornwood, NY).

### Real Time RT-PCR
Total RNA was isolated with the RNeasy plus mini kit (Qiagen; Germantown, MD). Reverse transcription of 1 μg of RNA in a total volume of 20 μl was performed with iScript cDNA Synthesis Kit (Bio-Rad) prior to real time PCR (iCycler; Bio-Rad, Hercules, CA). 25-μL amplification reactions contained primers (0.5 μM), dNTPs (0.2 mM each), 0.03 units Taq polymerase, and SYBR-green (Molecular Probes, Eugene, OR) at 1:150,000. Aliquots of cDNA were diluted 5 to 5000-fold to generate relative standard curves to which sample cDNA was compared. Alp forward primer: 5′-AACCCAGACACAAGCATTCC-3′ and reverse primer: 5′-GCCTTTGAGGTTTTTGGTCA-3′; Osx Forward primer: 5′-CCTCTCGACCCGACTGCAGATC-3′ and Reverse primer: 5′-AGCTGCAAGCTCTCTGTAACCATGAC-3′; aP2 forward primer: 5′-CATC

AGCGTAAATGGGGATT-3′ and reverse primer: 5′-TCGACTTTCCA TCCCACTTC-3′; Adiponectin Forward primer: 5′-GCA GAG ATG GCA CTC CTG GA-3′ and Reverse primer: 5′-CCC TTC AGC TCC TGT CAT TCC-3′; 18S forward primer: 5′-GAACGTCTGCCCTATCAACT-3′ and 18S reverse primer: 5′-CCAAGATCCAACTACGAGCT-3′. Standards and samples were run in triplicate. PCR products were normalized to 18 S amplicons in the RT sample, and standardized on a dilution curve from RT sample.

## Immunoblot analysis

Fractionated proteins were loaded onto a 7%–10% poly- acrylamide gel for chromatography and transferred to polyvinylidene difluoride membrane. After blocking, primary antibody was applied overnight at 4 °C including antibodies against H3K9me3 (cat# AB176916, Abcam), H3K27me3 (cat# ab6002, Abcam), ACTL6A (Arp4) (cat# SAB4503429, Sigma-Aldrich), ALBP (aP2) (cat# XG-6174, ProSci), Adiponectin (cat# PA1-054, Invitrogen), Tubulin (Cat# 91495, Abcam); Secondary anti- body conjugated with horseradish peroxidase was detected with ECL plus chemiluminescence kit (Amersham Biosciences/GE Healthcare, Piscataway NJ). The images were acquired with an HP Scanjet and densitometry determined using NIH ImageJ, 1.37 v.

## Cell modulus

Cell modulus measurements were performed as in ref. 4. Briefly, force-displacement curves for intact MSC nuclei were acquired using a Bruker Dimension FastScan Bio AFM. MLCT-SPH-5um -DC-A probes were used to decrease variables in calculating moduli. MSCs were located using the AFM's optical microscope and engaged on using a minimal force set point (1–3 nN) to ensure contact while minimizing applied force and resultant deformation prior to testing. Ramps were performed over the approximate center of each nucleus for all sam- ples. After engaging on a selected nucleus to ensure probe/nucleus contact as described above, force curve ramping was performed at a rate of 2 μm/s over 2 μm total travel (1 μm approach, 1 μm retract). 5 replicate force-displacement curves with an indentation depth of at least ~500 nm were acquired and saved for each nucleus tested, with at least 3 s of rest between ramps. Cell modulus was determined using Hertz model and analyzed using Atomic J[57].

## Nuclear G/F-Actin assay

To assay actin, $10^7$ cells per condition were collected for fractionation of nuclear proteins and assay using "G-actin/F-actin Assay Kit" (Cytoskeleton, Inc., Denver CO). As per instructions, 100 μl LAS2 buffer was added to nuclear pellets for homogenization through aspiration in a 25-G syringe; lysates were incubated at 37 °C for 10 min then cen- trifuged at 350 $g$ for 5 min at RT. Supernatant, containing G-actin, was then collected after 100,000 $g$ at 37 °C for 1 h. F-actin depolymeriza- tion buffer of 100 μl is added to the pellet to extract F-actin.

## IF Intensity measurement with ImageJ

Using "Analyze→ Set Measurements" to set parameters. Highlight the nuclear outside edge and select "Analyze →Measure" to measure immunofluorescence intensity (intensity for total nuclear area). Next, go "Edit→ Selection→ Enlarge" to shrink measurement area by choosing shrinking number and then measure intensity (intensity for central area). Subtract average intensity of total area from average intensity of central area = average intensity of nuclear periphery. Comparing average intensity of nuclear periphery to that of total area will be "Ratio of IF Mean Intensity of Nuc Periphery vs Total".

## Alp assay

The cells were scraped into Alp lysis buffer (10 mM Tris-HCl, pH 8.0; 1 mM MgCl2; 0.5% Triton X-100 in PBS). The samples were sonicated for 5 min. The Alp activity was measured with Alkaline Phosphatase Assay Kit (cat# MAK447, Sigma).

## ATAC-seq and data analysis

ATAC DNA was prepared from 50,000 MSC cells using an ATAC-seq protocol (Omni-ATAC) and primers as used prior[58] with transposase from llumina Kit (cat# 20034197). ATAC DNA quantity was measured by fluorometry (Qubit Q33266) and fragment size was confirmed on an Agilent TapeStation Agilent 5067-5584 (high sensitivity tape) and 5067-5585 (high sensitivity reagents).

Paired-end read data from each sample was processed using the PEPATAC pipeline[59] using the bowtie2 aligner[60] and the MACS2 peak caller[61]. Consensus peaks for each experimental condition consisted of the union set of peaks across replicates. Differentially accessible regions were determined using DESeq2[62] using default parameters. Accessible regions with greater than a 2-fold change ($|\log2FC| > 1$) and an adjusted $p < 0.01$ were considered significant. PCA plots were gen- erated with the pcaPlot function in DESeq2 using the top 1000 most variable regions after performing a variance stabilizing transformation using the vst function in DESeq2.

## RNA-seq and data analysis

Three distinct preparations for each culture condition were prepared for RNA-seq, which was sent to Novogene.

Paired-end reads were to the GRCm38 (mm10) genome assembly and quantified with Salmon v1.2.1[63] using default parameters and with a recent version of GENCODE[62] gene annotations (Release vM25). Dif- ferential expression analyses were performed using DESeq2[62] using default parameters. Genes with greater than a 2-fold change ($|\log2FC| > 1$) and an adjusted $p < 0.01$ were considered significant. PCA plots were generated with the using the top 1000 most variable genes after performing a variance stabilizing transformation using the vst function in DESeq2.

## Statistical analysis

The results are expressed as means ± SEM. For comparisons, 1-way analysis of variance or $t$ test (GraphPad Prism). Experiments were replicated at least twice to assure reproducibility. Densitometry data, where given, were compiled from at least 3 separate experiments as noted. AFM experiments were repeated 3 times. $P < 0.05$ were con- sidered significant.

## Reporting summary

Further information on research design is available in the Nature Portfolio Reporting Summary linked to this article.

# Data availability

All ATAC-seq and RNA-seq data is available in the Gene Expression Omnibus (GEO) under accession GSE242945. Source data are provided with this paper.

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

## Acknowledgements

Supported by NIAMS AR075803 (JR), AR073264 (MS), NIA AG069923 (GU), NSF 2025505 (GU).

## Author contributions

Conceptualization, B.S., G.U., T.F., and J.R.; Methodology, B.S., S.G.P., S.H., G.U., T.S.F., J.R.; Formal Analysis, B.S., Z.X., G.U., T.F., J.R.; Investigation, B.S., Z.X., M.D.T., C.M., G.U., T.F.; Writing – Original Draft, B.S., G.U., T.F., J.R., M.S.; Writing – Review & Editing, all authors; Funding Acquisition, J.R.

## Competing interests

The authors declare no competing interests.
