## [Peer Review File · Nature Communications]

Nuclear actin structure regulates chromatin accessibilityReviewers' Comments:

Reviewer #1:

Remarks to the Author:

In this research, the argument regarding nuclear actin changes is based primarily on the images that the authors were able to acquire. However, the authors mentioned the difficulty in visualizing F-actin within the nucleus due to the density of the nucleus. Without addressing this issue, conclusions regarding nuclear actin changes based on the current images are not sufficiently supported.

Further, the lack of key experimental details, such as sample size, number of replicates, and details of the measurement techniques of the assays performed, making it difficult to properly evaluate the data as well as the scope, reproducibility, and precision of the results.

The authors mainly discuss changes in F-actin vs G-actin but do not visualize intranuclear actin structures. For the western blot, without quantification, readers cannot determine from the images alone if there are actual differences in protein levels.

Whether there is a way to ensure the effects observed are specifically due to alterations in nuclear actin and not off-target effects of the inhibitors.

Whether there are changes in the localization of other actin-associated proteins (e.g. cofilin, profilin) under these conditions.

Whether there is potential feedback regulation of actin dynamics by chromatin and gene expression changes.

What would be changes in nuclear actin levels and structures over longer timescales during differentiation?

How do the observed effects vary across different cell types? The study only examines mesenchymal stem cells. Please provide information regarding whether those effects differ in MSCs from different sources and other cell types.

For the following questions, the authors mainly show observations but do not provide mechanistic insights:

How specifically do actin structures within the nucleus modulate chromatin organization and gene expression? What molecular mechanisms are involved?

How do alterations in nuclear actin levels and structures affect gene expression dynamics over longer timescales during differentiation? The authors only examine early time points.

How do changes in nuclear actin and chromatin organization feed back to regulate actin dynamics? The authors only examine effects of actin modifications on chromatin and gene expression.

The authors only show that Arp4 knockdown has effects but do not explain how Arp4 functionally contributes. How does Arp4 binding to actin monomers regulate gene transcription and chromatin accessibility.

Minors:

In the Methods section, the authors describe measuring the modulus of the cell nucleus using atomic force microscopy. However, in the Results section, you refer to this value as the 'cell modulus'. The nucleus modulus and cell modulus represent different mechanical properties, so using 'cell modulus' to describe the nucleus modulus measurements is misleading. I recommend revising the manuscript to consistently and correctly refer to your measurements and results as the 'nucleus modulus'.

Pay attention to typing mistakes: such as the unit of scale bar of Fig 1, should be " μm " not " μM ". Provide scale bars are missing in Fig 2, 5, and 6.

Reviewer #2:

Remarks to the Author:

In the reported work, Sen B et al investigated the role of nuclear actin structure in regulating chromatin accessibility and subsequently lineage-specific differentiation of MSCs. Using the Arp2/3 inhibitor CK666, the authors showed that disruption of nuclear actin structure significantly altered

chromatin access as measured with ATACseq, which was different from those caused by the actin polymerization inhibitor cytochalasin D (CytoD) although CytoD can enhance nuclear actin structure. Furthermore, Arp2/3 inhibition decreased pericentric H3K9me3 marks, whereas CytoD induced redistribution of H3K27me3 marks centrally. The authors demonstrated that the altered chromatin landscape impacted differential gene expression associated with osteogenic or adipogenic differentiation pattern. Lastly, the authors showed that silencing Arp4 led to extensive chromatin unpacking and gene accessibility and transcription. Thus, the authors concluded that dynamic actin remodeling may regulate chromatin interactions and lineage-specific differentiation in MSCs.

The reported findings are interesting and provide certain novel insights into the role of nuclear actin structure in regulating gene expression and stem cell differentiation. The work represents a logic extension of the authors' prior work on actin structure. The research design and experimental approaches were fairly straightforward. Data analysis and interpretations are appropriate. The overall conclusion is in general supported by the authors' experimental findings. However, the manuscript has apparent deficiencies and the following concerns need to be fully addressed:

- 1). While it is highly conceivable the actin-chromatin interactions would impact MSC differentiation (in this case, osteogenic and/or adipogenic differentiation), it is not clear whether nuclear actin structure alterations are cause or consequence of the differentiation process. The use of Arp2/3 inhibitor CK666 is insightful, but hardly physiological. It would be interesting to determine the status of the nuclear actin structure in MSCs at basal/default condition, and under osteogenic and/or adipogenic stimuli.
- 2). Since the inhibition of actin polymerization and secondary branching would impact chromatin access and subsequently transcription, will the promotion of actin polymerization (e.g., by overexpressing cofilin) cause the opposite effect on chromatin unpacking and transcription? If so, how would such change impact the differentiation of MSCs?
- 3). Since tubulins/microtubules are a critical part of cytoskeleton, would any efforts to stabilize or depolymerize tubulins/microtubules impact chromatin packing and gene transcription, compared with that for nuclear actin?
- 4). As the authors stated that the actin remodeling process is rather dynamic, it was not clearly justified why the ATACseq analysis was carried out at 24 h. It would be interesting to determine the chromatin unpacking status at earlier (e.g., 2h or 12h).
- 5). Along the same line, it would be interesting to determine H3K27me3 mark changes at earlier time points, and/or to determine those changes are correlated with chromatin unpacking and transcription.
- 6). Minor comments on wording and writing styles:
 - a). the title is somewhat ambiguous and does not reflect the effect of nuclear actin remodeling of stem cell differentiation.
 - b). The authors tended to use numerous ambiguous/unassertive statements in the manuscript: words like "alter" "alteration" "modified" are frequently used, which should be described in a more assertive fashion using "inhibit" "promote" "increase" or "decrease", etc.

Reviewer #3:

Remarks to the Author:

In this paper, Sen et al. investigated the contribution of nuclear actin structure to chromatin accessibility and gene expression using mesenchymal stem cells (MSC). This research group has previously shown the roles of nuclear actin in differentiation of MSC; CK666, an Arp2/3 inhibitor, and cytochalasin D (CytoD) differentially regulate MSC differentiation into osteoblasts or adipocytes by altering nuclear actin dynamics (Sen et al., Stem Cells 2015, 2017). In this manuscript, they took

advantage of this nuclear actin-regulated MSC differentiation system to ask if intranuclear actin structures affect gene accessibility. Authors found that the treatment of MSC with CK666 altered chromatin accessibility and H3K9me3 localization, while treatment with CytoD induced the redistribution of H3K27me3. In the meantime, global gene expression patterns were also affected by those drug treatments. Especially, genes related to the cell cycle and cellular proliferation were downregulated, reminiscent of differentiation away from the self-renewing MSC state. Furthermore, knockdown of Arp4 impaired differentiation of MSC and induced global opening of chromatin.

This paper highlights the importance of nuclear actin dynamics on MSC differentiation possibly through the regulation of chromatin structure and gene expression. However, with the current data shown in the manuscript, it is difficult to conclude that nuclear actin structure regulates chromatin accessibility (see major comments below).

Roles of nuclear actin dynamics in chromatin decondensation have already been shown (for example, in Baarlink et al., *Nat Cell Biol.* 2017 19(12):1389-1399; Krippner et al., *EMBO Rep.* 2020 21(11):e50758; Wang et al., *Nat Commun.* 2019 10(1):5271). The importance of nuclear actin in the establishment of histone modifications and chromatin organization has been proven by the Percipalle group (for example, in Mahmood et al., *Nat Commun.* 2021 12(1):5240). The necessity of Arp4 for MSC differentiation via chromatin regulation is an interesting finding, but this needs to be further investigated (see below).

Major comments;

1. The main conclusion of this paper relies on the fact that CK666 and CytoD alter nuclear actin dynamics in MSC. However, Figure 1 does not satisfactorily characterize nuclear actin dynamics after the treatments with CK666 and CytoD. Fig 1A and 1B do not show clear changes in nuclear F-actin staining. Quantification of nuclear F-actin needs to be performed and better images with clear nuclear phalloidin signals are necessary. Additionally, some statements are not well supported by the corresponding figures. For example, "CytoD, which fails to enter the nucleus due to lack of endogenous transporter mechanisms, promotes both actin transport into the nucleus and an increase in nuclear F-actin, ... (Fig. 1A)" is difficult to discern from Fig. 1A. Furthermore, references to relevant figures are not properly quoted at several sentences. For example, the first Fig. 1A in the Result section should be Fig. 1A and 1B, and the second Fig. 1B should be Fig. 1C.

Finally, authors need to characterize nuclear actin dynamics using the widely-used nuclear actin probe such as nuclear actin chromobody on top of the phalloidin staining.

2. Causative relationship between nuclear actin dynamics and changes in chromatin accessibility and gene expression is obscure. CK666 and CytoD influence both nuclear and cytoplasmic actin dynamics, and thus many cellular processes are disturbed by the treatment with these inhibitors. Further evidence is essential to support the idea that nuclear actin structure regulates chromatin accessibility. For example, specific knockout (knockdown) of Arp2/3 and increases/decreases of nuclear actin levels by ActinR62D and/or XPO6 should be tested.

3. Immunofluorescence analyses showed relocation of H3K9me3 and H3K27me3 after treatments with CK666 and CytoD, respectively. What is the functional role of the redistribution of histone marks?

4. One of the most significant findings in this manuscript is the role of Arp4 in MSC differentiation. More mechanistic insight into the Arp4-mediated control of MSC differentiation needs to be investigated. Did you find the enhanced nuclear F-actin assembly after the knockdown of Arp4 by phalloidin and chromobody staining? Was the chromatin opening after knockdown of Arp4 attributed to the increased nuclear actin assembly? Detailed investigation on the Arp4-mediated pathway will improve the quality and novelty of the paper.

Minor comments;

1. Throughout the manuscript, mis-citation to the figures are seen.
Ex); Arp3 remains both intra- and extra-nuclear, its location undisturbed by inhibition (Fig 1B).→this should be Fig. 1C
...K9-heterochromatin marked foci in a pattern similar to Fig 5A (Fig 6A,B). →this should be Fig. 6B
2. CK666 and CytoD treatments resulted in differential nuclear actin dynamics, but many common regions and genes were affected by these treatments (Fig. 3D and 4D). How do you explain this?
3. "Importantly, although both CK666 and CytoD increase the amount of nuclear actin (see Fig 1D)";
Fig 1D does not show the total amount of nuclear actin. Quantification is needed.
4. Statistical analysis needs to be done for Fig. 6C.

Reviewer #1

1. In this research, the argument regarding nuclear actin changes is based primarily on the images that the authors were able to acquire. However, the authors mentioned the difficulty in visualizing F-actin within the nucleus due to the density of the nucleus. Without addressing this issue, conclusions regarding nuclear actin changes based on the current images are not sufficiently supported.

We thank our reviewers for urging us to use the Nuclear Actin Chromobody-TagGFP plasmid to clarify nuclear actin structure results. In **new Fig 1C** and **S1b**, control cells and those treated with CK666 or CytoD after transfection with the NA-chromobody are shown 4 hours after treatment. NA-chromobody staining is dense in control cells with evidence of actin fibrils, consistent with phalloidin stain and Airyscan images (**Fig 1A,B**). Both NA-chromobody and phalloidin signals are less distinct in CK666 treated cells. In the CytoD condition, F-actin fibrils are clearly visible in the nucleus.

2. Further, the lack of key experimental details, such as sample size, number of replicates, and details of the measurement techniques of the assays performed, making it difficult to properly evaluate the data as well as the scope, reproducibility, and precision of the results.

We added experimental details to convey the following: high resolution images of cells were made from each of 6 wells. It was not possible to perform high resolution *nuclear* scans of multiple cells per well as the microscope processing time after zooming to capture images of a single nucleus leads to bleaching; as such, a single cell was chosen *at random* from each well of similar cells (6 wells /experiment) to image a nucleus in a high power field. To assure readers that our 6 imaged cells were not chosen for specific characteristics, we now include lower magnification images of multiple cells in a low resolution image which affords a clear picture that the pharmacologic agents affect *each* cell in the dish (**new Fig S1a**): Arp2/3 inhibition by CK666 causes a consistent clearing of actin connectivity to the nucleus (which we have previously described leads to increased nuclear height, Sen, 2017, ref #8). In contrast, CytoD treated cells have decreased F-actin polymers in the cytoplasm, but increased F-actin polymers within the nucleus.

The Results section includes:

*"...The findings shown throughout have been repeated in at least 3x 6-well experiments with a high power image from each experimental well examined. **Fig S1a** shows examples of the fields from which cells are selected in a non-biased fashion; cells representing each treatment condition (control, CK666, CytoD) are easily identifiable by the cellular actin staining pattern."*

The Methods section includes:

*"...For all experiments a single cell per HPF was imaged, to include 6 cells from each condition. The selection of the cell was random, any from a field which was significantly changed by the addition of either CK666, CytoD (**Fig S1a**) and conditions with siRNA added."*

3. The authors mainly discuss changes in F-actin vs G-actin but do not visualize intranuclear actin structures. For the western blot, without quantification, readers cannot determine from the images alone if there are actual differences in protein levels.

We thank the reviewer for this comment and agree. We now show new images of cells transfected with the nuclear actin chromobody (**Fig 1C**) to support phalloidin images as discussed in Q#1 and #2 above. We further added replicates to our measurements of extracted G- and F- nuclear actin in a total of 5 separate extraction experiments, allowing us to add densitometry data as a **new** graph in **Fig 1E**. As shown, nuclear F-actin statistically decreases in cultures treated with CK666 and increases with CytoD treatment.

4. The reviewer asks “Whether there are changes in the localization of other actin-associated proteins (e.g. cofilin, profilin) under these conditions. Whether there is potential feedback regulation of actin dynamics by chromatin and gene expression changes.”

We appreciate the reviewer’s inquiry. We have published data with knocked out of cofilin, profilin, importin-9 and exportin-6 (Sen et al, 2015, ref #5), showing that osteoblastogenesis in response to cytochalasin D requires the presence of both cofilin and importin-9, and that knock out of exportin and profilin do not alter this effect. Furthermore, we now cite others that demonstrated that KO of cofilin decreases the amount of nuclear actin, limiting substrate for polymerization (ref #51). As such, multiple actin associated proteins involved in nuclear import of actin will affect nuclear actin state in a dynamic fashion. We have added a new paragraph in the discussion with these lines page 14:

“...As actin structure within the nucleus is dependent both on transport into the nucleus requiring cofilin and importin-9^{5,51}, and can be enacted upon by an actin toolbox within the nucleus⁵², the dynamic control of nuclear actin level and structure is certain to be a central mechanism in modifying the epigenetic landscape and subsequent gene expression.”

5. What would be changes in nuclear actin levels and structures over longer timescales during differentiation?

Thank you for this interesting question: nuclear actin structures change rapidly and transiently in response to calcium, or in response to meiosis, as shown by many fine papers (added references #16,34,35,) and likely change continuously in response to the local microenvironment. In **new Fig S4** we show that under typical conditions enforcing phenotypic change in MSC (adipogenic medium x 3 days and osteogenic medium x 5 days), there are no readily visible changes in nuclear actin structure as examined with confocal microscopy despite clear changes in cytoplasmic actin structure, perhaps due to equilibration over the longer time period. We hope, in the future, to be able to follow labelled nuclear actin over time during various treatments with the help of a collaborator who has access to time-lapse fluorescent microscopy.

6. How do the observed effects vary across different cell types? The study only examines mesenchymal stem cells. Please provide information regarding whether those effects differ in MSCs from different sources and other cell types.

We thank the reviewer for this comment and address this question by measuring ATAC-seq at 4 hours in both the murine mdMSC studied in our work, and in the commonly studied murine cell line NIH-3T3 cell used in multiple studies of nuclear actin (e.g., ref #34-35). This data appears as a **new supplement, Fig S6**. These data reinforce our results showing that the physical alteration in actin state impacts chromatin. Interestingly, NIH3T3 cells appear to respond with changes in chromatin accessibility more rapidly than mdMSC cells. Unsurprisingly, NIH3T3 cells are highly different in terms of their ATAC-seq profiles, both in baseline control cells and after addition of CK666. We discuss these new findings in the results section, p9:

*“To provide a contrast in a different cellular setting, we measured chromatin accessibility in the well-studied NIH-3T3 embryonic fibroblast cell line, previously used to investigate intranuclear actin structure^{34,35}. Here, we found that 4 hours after addition of CK666, treated and control cells were clearly separated by PCA (**Fig S5a**): chromatin accessibility was significantly altered in 8091 regions (4827 increased, 3264 decreased; **Fig S5b**). Comparing to MSCs at both 4 and 24 hours post CK666 treatment, most of these changes were unique to the NIH 3T3 cells (**Fig S5c**). PCA of MSCs and NIH*

3T3 cells together show that chromatin profiles at baseline are extremely different, as expected for functionally distinct cells; this variation far exceeded the variation induced by CK666 (Fig S5d). Interestingly, pathways enriched by genes near altered chromatin in NIH 3T3 cells largely overlapped those enriched in MSCs, though there was not a significant signal for adipogenesis. These data suggest that specific regions altered by inhibition of secondary actin branching depend on the baseline cell chromatin state, but that there may be common cellular processes affected by these changes.”

7. How do alterations in nuclear actin levels and structures affect gene expression dynamics over longer timescales during differentiation? The authors only examine early time points.

Although we are unable to answer this interesting integrative question due to limited resources, we do show actin structures of differentiated cells in a new Fig S4, as discussed above in answer to Q#5. These micrographs show that phenotypic outcomes are accompanying by cytoskeletal change across the whole of the cell. For instance, the adipogenic cell has little cytoplasmic actin structure, while the osteoblast has much. Nuclear actin does not appear to be significantly different in the high resolution confocal micrographs.

8/&. How specifically do actin structures within the nucleus modulate chromatin organization and gene expression? What molecular mechanisms are involved? How do changes in nuclear actin and chromatin organization feedback to regulate actin dynamics? The authors only examine effects of actin modifications on chromatin and gene expression. The authors only show that Arp4 knockdown has effects but do not explain how Arp4 functionally contributes. How does Arp4 binding to actin monomers regulate gene transcription and chromatin accessibility.

We hope that future work of ours and others will add to an accumulating knowledge base studying the fundamental role that actin contributes to epigenetic structure. As to Arp4, data in Fig 6 and 7 provide new insights into Arp4's role in the nucleus, both as a critical member of multiple chromatin modifying complexes, its role in binding G-actin, and newly here as necessary for transcription of accessible chromatin regions. Arp4 is further visualized in differentiated MSC where there are no obvious changes in Arp4 despite differentiation of mdMSC into adipocytes or osteoblasts: Arp4 is present throughout the nucleus, with highest signal closest to the inner nuclear membrane (new Fig S4). It will require new lines of inquiry to gain more insight into the role of this protein. We have added further comments regarding Arp4 in several places in the enhanced discussion.

Minors:

In the Methods section, the authors describe measuring the modulus of the cell nucleus using atomic force microscopy. However, in the Results section, you refer to this value as the 'cell modulus'. The nucleus modulus and cell modulus represent different mechanical properties, so using 'cell modulus' to describe the nucleus modulus measurements is misleading. I recommend revising the manuscript to consistently and correctly refer to your measurements and results as the 'nucleus modulus'.

Pay attention to typing mistakes: such as the unit of scale bar of Fig 1, should be “ μm ” not “ μM ”
Provide scale bars are missing in Fig 2, 5, and 6.

Thank you for these comments. It is correct that we measured “cell modulus”. AFM measurements were taken on top of the nucleus of intact cells. Cell modulus measured in this way is largely representative of the nuclear modulus – but the reviewer is correct that the best appellation is cell modulus. To confirm the changes in the “nuclear modulus” we also conducted a nuclear modulus measurement utilizing nuclei isolated following either control or CytoD (where cell modulus changed). These results agree with the cell modulus data indicating decreased nuclear modulus under CytoD treatment. We corrected figures as noted, and added scale bars.

MSCs were treated with either DMSO or CytoD before isolation of nuclei. Within 1 hour of isolation, live isolated nuclei were placed on a Poly-L-Lysine covered dish and nuclear stiffness measured via AFM. Nuclear modulus is decreased after CytoD. Experiments were repeated 3 times, n=30 nuclei/grp. P< 0.0001

Reviewer #2:

In the reported work, Sen B et al investigated the role of nuclear actin structure in regulating chromatin accessibility and subsequently lineage-specific differentiation of MSCs. Using the Arp2/3 inhibitor CK666, the authors showed that disruption of nuclear actin structure significantly altered chromatin access as measured with ATACseq, which was different from those caused by the actin polymerization inhibitor cytochalasin D (CytoD) although CytoD can enhance nuclear actin structure. Furthermore, Arp2/3 inhibition decreased pericentric H3K9me3 marks, whereas CytoD induced redistribution of H3K27me3 marks centrally. The authors demonstrated that the altered chromatin landscape impacted differential gene expression associated with osteogenic or adipogenic differentiation pattern. Lastly, the authors showed that silencing Arp4 led to extensive chromatin unpacking and gene accessibility and transcription. Thus, the authors concluded that dynamic actin remodeling may regulate chromatin interactions and lineage-specific differentiation in MSCs.

The reported findings are interesting and provide certain novel insights into the role of nuclear actin structure in regulating gene expression and stem cell differentiation. The work represents a logic extension of the authors’ prior work on actin structure. The research design and experimental approaches were fairly straightforward. Data analysis and interpretations are appropriate. The overall conclusion is in general supported by the authors’ experimental findings.

Thank you for your interest in our work! We hope that the fundamental role that nuclear actin has in dynamic control of the epigenetic landscape will be studied in many laboratories in the coming years.

1). While it is highly conceivable the actin-chromatin interactions would impact MSC differentiation (in this case, osteogenic and/or adipogenic differentiation), it is not clear whether nuclear actin structure alterations are cause or consequence of the differentiation process. The use of Arp2/3 inhibitor CK666 is insightful, but hardly physiological. It would be interesting to determine the status of the nuclear actin structure in MSCs at basal/default condition, and under osteogenic and/or adipogenic stimuli.

We thank the reviewer for this comment and have added a new Fig S4a,b to respond to this question. Here we have used standard methodology to achieve osteogenesis and adipogenesis. Actin structure throughout the cells in these 2 differentiated states by itself allows distinction of cell type from control. The osteoblast has more total actin structure, and the adipocyte less. The nuclei do not show visible changes in actin structure. This would suggest that changes in chromatin structure due to actin remodeling are initiating events, rather than consequences. We have mentioned this in the text, page 8:

“... Shown in Fig S4, where standard medias were used to terminally differentiate osteoblasts and adipocytes⁵, different actin cytoskeletons sort to these phenotypes: actin structure is depleted in

adipocytes and increased in osteoblasts. Nuclear actin and Arp4 staining are not visibly different in any of the differentiated nuclei compared with control cells. This suggests that significant changes in nuclear actin and chromatin states caused by actin remodeling drugs, while initiating differentiating events, eventually achieve some adaptive equilibrium.”

2). Since the inhibition of actin polymerization and secondary branching would impact chromatin access and subsequently transcription, will the promotion of actin polymerization (e.g., by overexpressing cofilin) cause the opposite effect on chromatin unpacking and transcription? If so, how would such change impact the differentiation of MSCs?

Thank you for this question: we have increased actin polymerization with the use of CytoD, which drives actin into the nucleus and increases polymerization (*revised Fig 1E*) as well as induces osteogenesis. We previously published regarding cofilin (and importin-9) showing that knockout of either co-transporter prevents CytoD's differentiative effect by limiting the transport of actin into the nucleus (Sen et al, 2015, ref #5). Although we have not overexpressed cofilin, it would be predicted to alter actin structure not only in the nucleus but throughout the cell by virtue of its role in polymerization dynamics and nuclear transport.

3). Since tubulins/microtubules are a critical part of cytoskeleton, would any efforts to stabilize or depolymerize tubulins/microtubules impact chromatin packing and gene transcription, compared with that for nuclear actin?

Thank you for this interesting question as to how our findings fit in the wider context of dynamic cytoskeletal structure. We added a comment to a new paragraph in the discussion, p 14:

“... The story certainly extends beyond the nucleus to cytoplasmic actin structure which apply force on the nucleus⁵⁰ as well as many other modifiable structural proteins (e.g., microtubules) that connect and interact with the LINC complex and thus to static structural proteins inside the nucleus (e.g., lamins).”

4). As the authors stated that the actin remodeling process is rather dynamic, it was not clearly justified why the ATACseq analysis was carried out at 24 h. It would be interesting to determine the chromatin unpacking status at earlier (e.g., 2h or 12h).

Thank you for this excellent suggestion. We have added 2 additional ATACseq data sets of both mdMSC and NIH3T3 cells at 4 hours (*new Figs S5,6*). The mdMSC data is quite interesting as it shows that there are less accessible chromatin regions affected at 4 hours than at 24 hours, but they largely involve the same pathways. This stimulates an idea that the chromatin unpacking/packing due to actin remodeling is not random! The substantially enriched results section refers to this new data beginning on p 9:

“We also examined CK666 induced changes in chromatin accessibility at 4 h to complement the 24 hour data set. We found that while changes in chromatin accessibility again clearly separated CK666 samples from control in a PCA (Fig S5a), there were fewer altered regions (1962 total, 1098 increased, 864 decreased; Fig S5b). Of these, 774 were shared at 4 hours and 24 hours (Fig S5c), with associated genes enriched in common pathways, notably adipogenesis (Fig S5d), suggestive of a progressive mechanism whereby chromatin is modified.

To provide a contrast in a different cellular setting, we measured chromatin accessibility in the well-studied NIH-3T3 embryonic fibroblast cell line, previously used to investigate intranuclear actin

structure 34, 35. Here, we found that 4 hours after addition of CK666, treated and control cells were clearly separated by PCA (Fig S6a): chromatin accessibility was significantly altered in 8091 regions (4827 increased, 3264 decreased; Fig S6b). Comparing to MSCs at both 4 and 24 hours post CK666 treatment, most of these changes were unique to the NIH 3T3 cells (Fig S6c). PCA of MSCs and NIH 3T3 cells together show that chromatin profiles at baseline are extremely different, as expected for functionally distinct cells; this variation far exceeded the variation induced by CK666 (Fig S6d). Interestingly, pathways enriched by genes near altered chromatin in NIH 3T3 cells largely overlapped those enriched in MSCs, though there was not a significant signal for adipogenesis. These data suggest that specific regions altered by inhibition of secondary actin branching depend on the baseline cell chromatin state, but that there may be common cellular processes affected by these changes.”

5). Along the same line, it would be interesting to determine H3K27me3 mark changes at earlier time points, and/or to determine those changes are correlated with chromatin unpacking and transcription.

We have utilized histone marks to confirm that there are structural changes in the epigenetic landscape. To correlate the alterations in geography with chromatin unpacking/transcription is, we hope the reviewer will agree, outside of the scope of the work presented here.

6). Minor comments on wording and writing styles:

a). the title is somewhat ambiguous and does not reflect the effect of nuclear actin remodeling of stem cell differentiation.

b). The authors tended to use numerous ambiguous/unassertive statements in the manuscript: words like “alter” “alteration” “modified” are frequently used, which should be described in a more assertive fashion using “inhibit” “promote” “increase” or “decrease”, etc.

Thank you for this comment; when possible, we have used more assertive terms. However, we intend the language to accurately reflect the findings. It is clear, for example that chromatin accessibility decreases and increases in nearly equal amounts – thus ‘modification’ and ‘alteration’ are often the most appropriate words. We further hesitate to change the title as we study changes in nuclear actin that lead to either adipogenesis (CK666) or osteoblastogenesis (CytoD). Our novel results represent an unbiased examination of how these actin modifiers change chromatin accessibility and gene expression at early stages in the progress *towards* two differentiated states.

Reviewer #3:

In this paper, Sen et al. investigated the contribution of nuclear actin structure to chromatin accessibility and gene expression using mesenchymal stem cells (MSC). This research group has previously shown the roles of nuclear actin in differentiation of MSC; CK666, an Arp2/3 inhibitor, and cytochalasin D (CytoD) differentially regulate MSC differentiation into osteoblasts or adipocytes by altering nuclear actin dynamics (Sen et al., Stem Cells 2015, 2017). In this manuscript, they took advantage of this nuclear actin-regulated MSC differentiation system to ask if intranuclear actin structures affect gene accessibility. Authors found that the treatment of MSC with CK666 altered chromatin accessibility and H3K9me3 localization, while treatment with CytoD induced the redistribution of H3K27me3. In the meantime, global gene expression patterns were also affected by those drug treatments. Especially, genes related to the cell cycle and cellular proliferation were downregulated, reminiscent of differentiation away from the self-renewing MSC state. Furthermore, knockdown of Arp4 impaired differentiation of MSC and induced global opening of chromatin.

This paper highlights the importance of nuclear actin dynamics on MSC differentiation possibly through the regulation of chromatin structure and gene expression. However, with the current data shown in the manuscript, it is difficult to conclude that nuclear actin structure regulates chromatin accessibility (see major comments below).

Roles of nuclear actin dynamics in chromatin decondensation have already been shown (for example, in Baarlink et al., Nat Cell Biol. 2017 19(12):1389-1399; Krippner et al., EMBO Rep. 2020 21(11):e50758; Wang et al., Nat Commun. 2019 10(1):5271). The importance of nuclear actin in the establishment of histone modifications and chromatin organization has been proven by the Percipalle group (for example, in Mahmood et al., Nat Commun. 2021 12(1):5240). The necessity of Arp4 for MSC differentiation via chromatin regulation is an interesting finding, but this needs to be further investigated (see below).

We are grateful for R3's interest and expertise in the area of nuclear actin. We contend that our data is not iterative of the many papers cited above, which are now referenced in the discussion. Importantly these three papers show *transient* changes in nuclear actin, while we examine in detail chromatin accessibility and gene expression after imposing *long term* changes in nuclear actin state. With regard to Mahmood et.al., referenced in the original version of our paper, actin was knocked out, thus actin structure was not under consideration. Here our studies using ATAC-seq and RNAseq to identify multiple changes in chromatin accessibility and gene expression show unique responses to changes in structural actin. Chromatin accessibility and gene transcription, as we show, are preceded by alterations in location of histone marks within the nucleus. Further, we believe that our paper is qualitatively different in that it examines early states in cells that differentiate down completely different pathways, and quantitatively identifies gene pathways associated with the progressive differentiation.

In response, we added a new paragraph in the Discussion to better incorporate the context of our work within the literature (p 14):

“Previous work has evaluated transient changes in nuclear actin state and chromatin geography. F-actin fibers form during mitotic exit, a process which involves the interacting molecule cofilin-1 that both severs actin filaments and transports actin monomers into the nucleus¹⁶. The appearance of F-actin invokes chromatin decondensation and nuclear expansion. Reorganization of chromatin after mitosis requires the presence of α -actinin 4, the full process occurring over about 60 minutes³⁴. Interestingly, changes in intracellular calcium, either due to treatment with an ionophore, or through G-protein coupled receptors, can induce rapid stimulation of nuclear actin polymerization that peaks at less than 2 minutes before returning to a basal state³⁵. And, as noted above, knocking out nuclear β -actin, essentially decreasing any actin contribution to structure, leads to widespread changes in chromatin arrangement⁷. Our data complements and adds detail to the types of nuclear actin structure that can lead to changes in specific chromatin accessibility in both MSCs and NIH3T3 cells. The story certainly extends beyond the nucleus to cytoplasmic actin structure which apply force on the nucleus⁵⁰ as well as many other modifiable structural proteins (e.g., microtubules) that connect and interact with the LINC complex and thus to static structural proteins inside the nucleus (e.g., lamins). As actin structure within the nucleus is dependent both on transport into the nucleus requiring cofilin and importin-9^{5,51}, and can be enacted upon by an actin toolbox within the nucleus⁵², the dynamic control of nuclear actin level and structure is certain to be a central mechanism in modifying the epigenetic landscape and subsequent gene expression.”

Major comments

1. The main conclusion of this paper relies on the fact that CK666 and CytoD alter nuclear actin dynamics

in MSC. However, Figure 1 does not satisfactorily characterize nuclear actin dynamics after the treatments with CK666 and CytoD. Fig 1A and 1B do not show clear changes in nuclear F-actin staining. Quantification of nuclear F-actin needs to be performed and better images with clear nuclear phalloidin signals are necessary. Additionally, some statements are not well supported by the corresponding figures. For example, “CytoD, which fails to enter the nucleus due to lack of endogenous transporter mechanisms, promotes both actin transport into the nucleus and an increase in nuclear F-actin, (Fig. 1A)” is difficult to discern from Fig. 1A. Furthermore, references to relevant figures are not properly quoted at several sentences. For example, the first Fig. 1A in the Result section should be Fig. 1A and 1B, and the second Fig. 1B should be Fig. 1C. Finally, authors need to characterize nuclear actin dynamics using the widely-used nuclear actin probe such as nuclear actin chromobody on top of the phalloidin staining.

Thank you for this important comment which we greatly appreciate. To respond we performed new studies with nuclear actin chromobody. Please see Q#1 to R1 above. In sum, we provide clear NA-chromobody visualization showing decreased NA chromobody signal compared to control cells in those treated with CK666, and increased fibrils in those treated with CytoD (**new Fig 1C**).

Regarding G/F actin measurements, we have now performed nuclear extractions in 5 separate experiments to be able to provide densitometry in the **new quantitative panel of Fig 1E**. As shown, nuclear F-actin decreases in cultures treated with CK666 and increases with CytoD treatment.

We addressed errors in figure labeling and carefully checked to ensure correctness in this revision.

2. Causative relationship between nuclear actin dynamics and changes in chromatin accessibility and gene expression is obscure. CK666 and CytoD influence both nuclear and cytoplasmic actin dynamics, and thus many cellular processes are disturbed by the treatment with these inhibitors. Further evidence is essential to support the idea that nuclear actin structure regulates chromatin accessibility. For example, specific knockout (knockdown) of Arp2/3 and increases/decreases of nuclear actin levels by ActinR62D and/or XPO6 should be tested.

We have previously knocked out Arp2/3, but were unable to explore much as it leads to cell death in our MSC line (ref #8). In answer to R1 Q#4 we discuss that we have published that nuclear actin export (knocked out exportin-6, profilin) did not alter CytoD induction of osteogenesis, which contrasted with knockout of either cofilin or importin-9, co-transporters of actin into the nucleus, which prevented osteogenesis. This indicates that actin related proteins responsible for transport of actin into the nucleus, causing increased nuclear actin, will affect the epigenetic landscape. We have added these concepts to a paragraph in the discussion, p 14, which ends:

“...As actin structure within the nucleus is dependent both on transport into the nucleus requiring cofilin and importin-9^{5,51}, and can be enacted upon by an actin toolbox within the nucleus⁵², the dynamic control of nuclear actin level and structure is certain to be a central mechanism in modifying the epigenetic landscape and subsequent gene expression.”

3. Immunofluorescence analyses showed relocation of H3K9me3 and H3K27me3 after treatments with CK666 and CytoD, respectively. What is the functional role of the redistribution of histone marks?

The goal in showing these histone marks was twofold. First we achieved a static visualization of changes in chromatin modifications that precede the change in chromatin accessibility. Secondly, we were able to show that CK666 and CytoD had significantly different effects on two chromatin marks at

an early time point. Our goal was to show that actin structure is accompanied by geographic changes in placement of histone marks which preceded the changes in chromatin accessibility measured by ATAC-seq.

4. One of the most significant findings in this manuscript is the role of Arp4 in MSC differentiation. More mechanistic insight into the Arp4-mediated control of MSC differentiation needs to be investigated. Did you find the enhanced nuclear F-actin assembly after the knockdown of Arp4 by phalloidin and chromobody staining? Was the chromatin opening after knockdown of Arp4 attributed to the increased nuclear actin assembly? Detailed investigation on the Arp4-mediated pathway will improve the quality and novelty of the paper.

Thank you for this suggestion and your interest. We knocked out Arp4 because of literature suggesting that it interacted with both G-actin and HP1 α in the periphery of the nucleus, leading us to think we might pinpoint a mechanism for why increased nuclear G-actin caused adipogenesis. Instead we found that Arp4 was essential for transcriptional control – without which adipogenesis could not occur. Arp4 is clearly involved in nearly every modification of chromatin. We agree that Arp4 is fascinating and requires additional, future investigation. We hope R3 can agree that a detailed study of Arp4 in MSC differentiation is beyond the scope of the novel work presented here.

Minor comments

1. Throughout the manuscript, mis-citation to the figures are seen. Ex); Arp3 remains both intra- and extra-nuclear, its location undisturbed by inhibition (Fig 1B). →this should be Fig. 1C...K9-heterochromatin marked foci in a pattern similar to Fig 5A (Fig 6A,B). →this should be Fig. 6B

We have revised the manuscript including references to figures within the text. Thank you for your close reading of the manuscript.

2. CK666 and CytoD treatments resulted in differential nuclear actin dynamics, but many common regions and genes were affected by these treatments (Fig. 3D and 4D). How do you explain this?

During pathway analysis, shared processes in cells that move along completely separate differentiative pathways can be critical for basic cell processes required for change. This is certainly highlighted in new data where we now report many of the same pathways rising to significance in an *entirely* different cell line, the NIH3T3 (**new Fig S6**). PCA analysis of NIH3T3 cells treated with CK666 shows complete separation from control cells – but both CK666 and control NIH3T3 cells are even more distinct from that of the mdMSC cell line than they are from each other. Shared pathway changes due to actin remodeling might suggest that there are specific loci that are more affected by actin structure, and these seem to be near genes common to particular pathways. The loci affected in a specific cell type (MSC) or condition (CK666, CytoD) will depend on the control chromatin state of that cell and how actin is changed, but a subset of these loci may be sensitive to any disruption of actin structure. We note that the common changes in chromatin accessibility (**Fig 3D**) involve <20% of the total in each cell type. As stated in the paper, this is proportionally less than the percentage of common changes in gene expression.

3. “Importantly, although both CK666 and CytoD increase the amount of nuclear actin (see Fig 1D)”; Fig 1D does not show the total amount of nuclear actin. Quantification is needed.

This is now quantified and revised see **new densitometry** in current **Fig 1E**.

4. Statistical analysis needs to be done for Fig. 6C.

Performed and added.

Reviewers' Comments:

Reviewer #1:

None

Reviewer #2:

Remarks to the Author:

The authors were responsive and addressed most if not all of the reviewers' concerns. The revised manuscript has been improved and is thus recommended for consideration for acceptance.

Reviewer #3:

Remarks to the Author:

The manuscript has been improved and the authors have satisfactorily answered the concerns.

Minor point:

P6. "...visualized there (Fig. 1C)" should be "...visualized there (Fig. 1D)".

NCOMMS-23-40272B

Response to REVIEWERS' COMMENTS

Reviewer #2 (Remarks to the Author):

The authors were responsive and addressed most if not all of the reviewers' concerns. The revised manuscript has been improved and is thus recommended for consideration for acceptance.

Thank you.

Reviewer #3 (Remarks to the Author):

The manuscript has been improved and the authors have satisfactorily answered the concerns.

Minor point:

P6. "...visualized there (Fig. 1C)" should be "...visualized there (Fig. 1D)".

Thank you, we have corrected the minor point.